# Comprehensive structural characterization of the human AAA+ disaggregase CLPB in the apo- and substrate-bound states reveals a unique mode of action driven by oligomerization

**Damu Wu[1], Yan Liu[2], Yuhao Dai[1,3], Guopeng Wang[1], Guoliang Lu[2], Yan Chen[1], Ningning Li[1,4], Jinzhong Lin[2]\*, Ning Gao[1,4,5]\***

**1** State Key Laboratory of Membrane Biology, Peking-Tsinghua Joint Center for Life Sciences, School of Life Sciences, Peking University, Beijing, China, **2** State Key Laboratory of Genetic Engineering, School of Life Sciences, Zhongshan Hospital, Fudan University, Shanghai, China, **3** Academy of Advanced Interdisciplinary Studies, Peking University, Beijing, China, **4** Changping Laboratory, Beijing, China, **5** National Biomedical Imaging Center, Peking University, Beijing, China

\* linjinzhong@fudan.edu.cn (JL); gaon@pku.edu.cn (NG)

**Data Availability Statement:** Data are available from the Protein Data bank and Electron Microscopy Data Bank.The cryo-EM density maps

## Abstract

The human AAA+ ATPase CLPB (SKD3) is a protein disaggregase in the mitochondrial intermembrane space (IMS) and functions to promote the solubilization of various mitochondrial proteins. Loss-of-function CLPB mutations are associated with a few human diseases with neutropenia and neurological disorders. Unlike canonical AAA+ proteins, CLPB contains a unique ankyrin repeat domain (ANK) at its N-terminus. How CLPB functions as a disaggregase and the role of its ANK domain are currently unclear. Herein, we report a comprehensive structural characterization of human CLPB in both the apo- and substrate-bound states. CLPB assembles into homo-tetradecamers in apo-state and is remodeled into homo-dodecamers upon substrate binding. Conserved pore-loops (PLs) on the ATPase domains form a spiral staircase to grip and translocate the substrate in a step-size of 2 amino acid residues. The ANK domain is not only responsible for maintaining the higher-order assembly but also essential for the disaggregase activity. Interactome analysis suggests that the ANK domain may directly interact with a variety of mitochondrial substrates. These results reveal unique properties of CLPB as a general disaggregase in mitochondria and highlight its potential as a target for the treatment of various mitochondria-related diseases.

## Introduction

Protein misfolding and aberrant aggregation are devastating to many fundamental functions of the cell and failures to remediate them are closed related to many human diseases [1,2]. To maintain a healthy proteome, cells have evolved multiple dedicated systems, one of which is

have been deposited in the Electron Microscopy Data Bank with the accession codes EMD-33105, EMD-33106, EMD-33109, EMD-33110 and EMD-33104 for the CLPB, CLPBE425Q, NBDE425Q-hexamer, NBDE425Qheptamer and NBDE425Q-nanomer, respectively. The atomic model of NBDE425Qnanomer has been deposited in Protein Data Bank with accession code PDB 7XBK.The coordinate and the structure factor of CLPB-ANK have been deposited in the Protein Data Bank with accession code 7XC5. All other relevant data are within the paper and its Supporting Information files.

**Funding:** The work was supported by the National Natural Science Foundation of China (https://www.nsfc.gov.cn/) (32230051 to N.G., 31922036 to N.L.), the National Key Research and Development Program of China (https://service.most.gov.cn/) (2019YFA0508904 to N.G.), and the Qidong-SLS Innovation Fund to N.G. This work is also partially supported by Changping Laboratory. The funders had no role in study design, data collection and analysis, decision to publish, or preparation of the manuscript.

**Competing interests:** The authors have declared that no competing interests exist.

**Abbreviations:** AM, ankyrin motif; AML, acute myeloid leukemia; CTF, contrast transfer function; FDR, false discovery rate; IMS, intermembrane space; LH, linker helix; MPP, mitochondrial processing peptidase; MS, mass spectrometry; MTS, mitochondrial targeting signal; NBD, nucleotide-binding domain; nsEM, negative staining electron microscopy; NTD, N-terminal domain; PARL, presenilin-associated rhomboid-like; PL, pore-loop; RT, room temperature; SCN, severe congenital neutropenia; WT, wild type; 3-MGA, 3-methylglutaconic aciduria.

the HSP100 chaperone family [3–5]. As a subfamily of the AAA+ ATPase, HSP100 proteins generally contain an N-terminal domain (NTD), 1 or 2 ATPase domains (or nucleotide-binding domain, NBD), and usually function in hexameric forms. Taking yeast Hsp104 as an example, the NTD is involved in substrate binding, while the NBD1 and NBD2 bind and hydrolyze ATP to facilitate power substrate unfolding and translocation [6]. Similar to many other AAA + ATPases, Hsp104 unfolds and transports substrates through its central pore by a ratchet-like motion of the highly conserved pore-loops (PLs) on the ATPase domains [7]. The ATP-hydrolysis cycle-dependent conformational change of each subunit results in both inter- and intra-subunit structural remodeling, which collectively lead to the threading and unidirectional movement of the peptide within the central channel [7].

Recently, a new type of HSP100 family proteins, CLPB (also known as SKD3) was reported to act as a protein disaggregase in the intermembrane space (IMS) of mitochondria [8–11]. The N-terminus of CLPB has a mitochondrial targeting signal (MTS), followed by an ankyrin repeat (ANK) domain, and ends with a C-terminal NBD (Fig 1A). There is a short hydrophobic stretch between the MTS and the first ankyrin motif (AM), as well as a long linker helix (LH) between ANK domain and NBD (Fig 1A). The MTS is cleaved by mitochondrial processing peptidase (MPP), followed by a second cleavage by presenilin-associated rhomboid-like (PARL) protease to remove additional hydrophobic residues from the N-terminus [11,12]. The ANK domain is a unique feature of CLPB compared to other AAA+ ATPases, and the removal of ANK domain disrupts the disaggregase activity of CLPB [8].

Dysfunction of CLPB by mutations is associated with several human diseases, such as the 3-methylglutaconic aciduria (3-MGA) [10,13–16]. A common disorder of the 3-MGA patients is the increased urinary 3-methylglutaric acid excretion, often with varying degree of microcephaly, small birth weight, neutropenia, severe encephalopathy, intellectual disability, movement disorder, and cataracts [10,13–16]. Moreover, heterozygous missense variants of CLPB were also identified in patients with severe congenital neutropenia (SCN), and these variants were found to disrupt granulocyte differentiation of human hematopoietic progenitors [17]. Many of these disease-related mutations have been shown to impair the disaggregase activity of CLPB, such as T268M, R408G, R475Q, A591V, R650P in 3-MGA and N499K, E557K, R561G, R620C in SCN [8,17]. At the cellular level, CLPB is important in maintaining normal cristae structure of mitochondria [18]. CLPB interacts with HAX1, an anti-apoptotic factor of BCL-2 family, to promote cell survival [10,18]. Up-regulated cellular level of CLPB in acute myeloid leukemia (AML) cells was found to mediate the resistance to BCL-2 inhibitor venetoclax [18]. Recently, CLPB was also found to have negative correlation with the progression-free survival in castration-resistant prostate cancer [19].

It is currently not clear why CLPB has such a broad role in different aspects of mitochondrial function and how the ANK and NBD domains work together to fulfill its essential disaggregase activity. Here, we present a structural and functional characterization of human CLPB. Unexpectedly, we found that CLPB assembles into a homo-tetradecamers in the absence of substrate. Upon substrate binding, the CLPB complex is converted into dodecamers consisting 2 conventional hexameric units. The NBD ring within a hexamer shares common structural features of typical AAA+ unfoldase/disaggregase, with a spiral arrangement of PLs to interact with a fully threaded substrate. The N-terminal ANK domain is essential for the higher-order organization of CLPB and contributes to the disaggregase activity by directly interacting with various mitochondrial substrates. These results provide a framework for further dissection of the role of CLPB in regulating various mitochondrial functions.

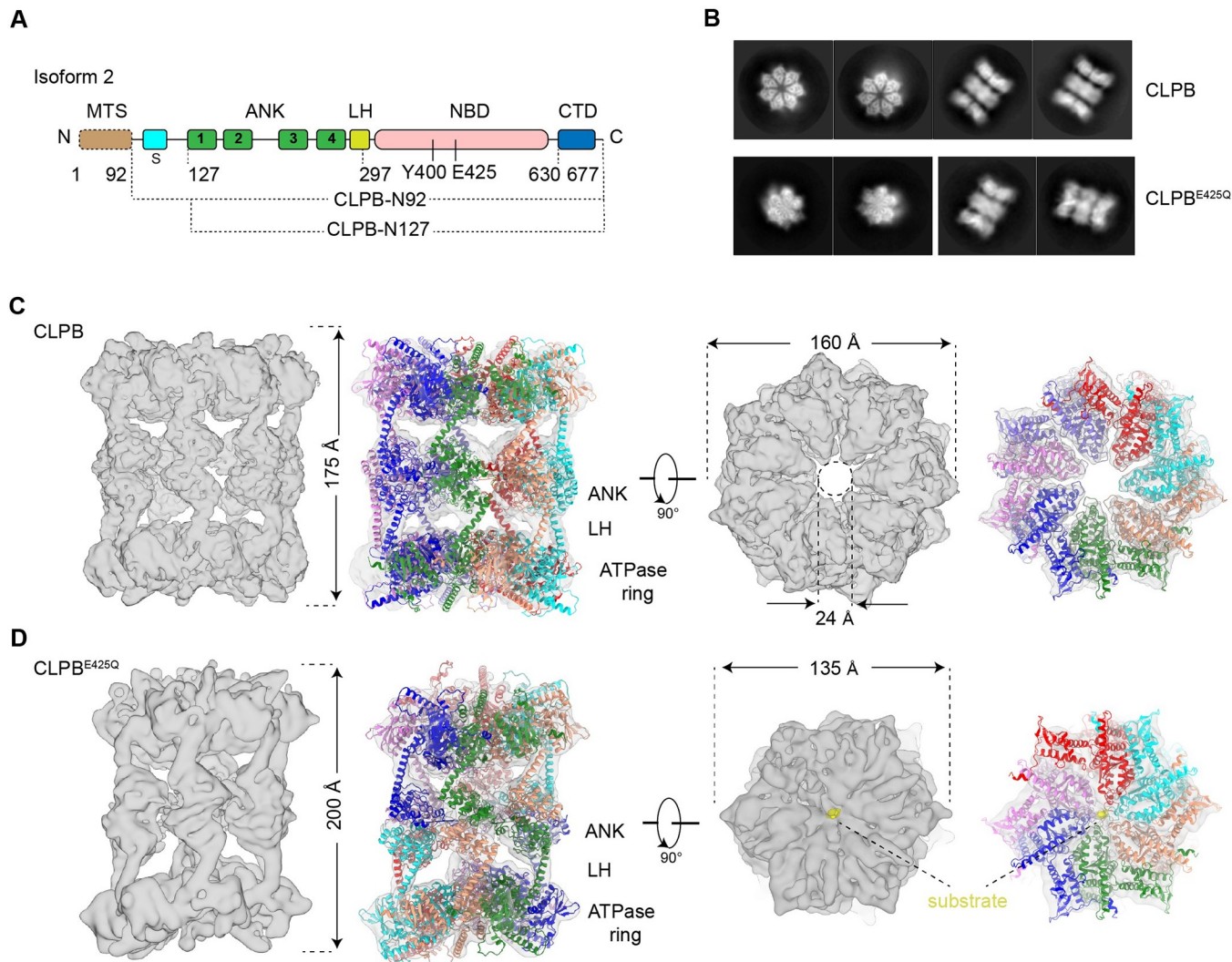

**Fig 1. Cryo-EM structures of the CLPB double-heptamer in the apo-state and the CLPB^E425Q double-hexamer in the substrate-bound state.** (**A**) Domain organization of *H. sapiens* CLPB. CLPB is composed of an MTS, a short hydrophobic stretch (S), an LH, 4 ankyrin-repeat (ANK) motifs, an NBD, and a CTD. (**B**) Representative 2D classification averages of CLPB and CLPB^E425Q datasets. (**C, D**) The density maps of the double-heptamer (C) and double-hexamer (D), superimposed with the models of CLPB. The density maps are shown in the side and top (ATPase ring) views. The higher-order oligomer is mediated by ANK domains. The substrate was labeled as yellow. LH, linker helix; NBD, nucleotide-binding domain.

## Results

### Cryo-EM structures of CLPB in the apo-state and substrate-bound state

We cloned the coding sequence of *CLPB* from a homemade cDNA library of HEK-293T cells, and sequencing result indicated that it was the splicing isoform 2 (UNIPROT: Q9H078-2). Different expression constructs were tested. Firstly, CLPB without the MTS [10], named CLPB-N92 (Fig 1A), was expressed and purified from *E. coli* cells. Consistent with a previous study [9], CLPB-N92 formed large aggregates and was highly heterogenous in size as shown by gel filtration and negative staining electron microscopy (nsEM) (S1A and S1B Fig). In addition to the signal peptide removal, CLPB was reported to be further processed by a mitochondrial protease PARL localized in the inner membrane, and the predicted cleavage site is between

C126 and Y127 [12]. Notably, a hydrophobic stretch within the region removed by PARL (Fig 1A) was reported to inhibit the disaggregase activity of CLPB [8]. Therefore, as a validation, we expressed full-length CLPB exogenously in HEK-293T cells and examined the sizes of protein products. Western blotting analysis showed 2 clear bands with the precursor form gradually decreasing over time (S1C Fig). Next, another construct harboring the sequence of CLPB starting from PARL cleavage site (named CLPB-N127) was tested for HEK-293T expression, which resulted in a protein product in the same size as the mature form of CLPB (S1D Fig). Therefore, the CLPB-N127 construct was finally used for CLPB protein preparation from *E. coli* cells.

Interestingly, both gel filtration and nsEM showed that the purified CLPB complex is in a higher oligomeric state rather than a hexamer as expected from typical AAA+ unfoldases and disaggregases (S2A and S2B Fig). Very recently, 2 independent studies have also reported that purified CLPB exhibits an unusual higher-order oligomeric form [20,21]. And this higher-order organization is irrelevant of exogenously supplemented AMPPNP (S2A Fig). Despite this unusual observation, the purified CLPB complex is competent in both ATPase and disaggregase activities (S1E and S1F Fig and S1 Data). AMPPNP-treated CLPB complexes were then subjected to cryo-EM analysis. The 2D classification showed a three-layered architecture and a heptameric feature for the side-view and top-view average images, respectively (Fig 1B). Further 3D classification indicated that CLPB complexes are double-heptamers, and they are extremely dynamic in structure, with both inter- and intra-heptamer conformational variations. With different 3D classification strategies, we could only push the overall resolution to a range of 6 to 7 Å (S3 Fig and S1 Table). At this resolution, secondary structures were resolved in certain regions, and the map matches well with the predicted model of CLPB by AlphaFold2 [22]. The overall size of the complex is 175 Å and 160 Å in height and width, respectively. The 7 NBDs form a closed ring, with an open channel (24 Å in diameter) in the center. The "head-to-head" organization of the 2 heptamers is mediated by their N-terminal ANK domains (Fig 1C). Although a heptameric form of the ATPase modules is rare among AAA+ ATPases, another mitochondrial AAA+ ATPase Bcs1 was recently reported to exist in a homo-heptamer form [23,24].

Since the double-heptamer contains no substrate and is widely open in the central pore, we set out to obtain a structure of the substrate-engaged CLPB complex. A convenient way of doing this is through a Walker B mutation (E425Q) in the NBD domain. For most of the AAA+ proteins, this mutation would greatly slow down ATP hydrolysis but not ATP binding. Structural studies of a few AAA+ unfoldases/disaggregases showed that this mutation often resulted in a co-purification of endogenous peptide in the central channel [25–28]. Therefore, we prepared CLPB$^{E425Q}$ mutant from *E. coli* cells (S2C and S2D Fig) and analyzed the sample by cryo-EM (S4 Fig). Unexpectedly, top-view class averages from 2D classification showed a hexameric ring (Figs 1B and S5B). Similar to the wild-type (WT) CLPB complex, the mutant complex is also highly dynamic and the global density map could only be refined to an overall resolution of 7.9 Å. With local refinement, the NBD ring was improved to 5.2 Å (S4C, S4D, and S4F Fig). From the structure, it is clear that the mutant CLPB$^{E425Q}$ now takes a double-hexamer (dodecamer) form. Compared with the WT structure, while the width of the mutant complex reduces from 160 Å to 135 Å, the overall height increases from 175 Å to 200 Å (Fig 1D). This is due the spiral configuration of CLPB$^{E425Q}$ subunits within each hexamer. From the improved map of the NBD ring, the substrate density in the central channel could be unambiguously identified (S4D Fig). In general, the helical arrangement of subunits within a hexamer is highly similar to other substrate-bound AAA+ ATPases [7,25,28–30].

To confirm that this higher-order organization of CLPB is not an artifact of exogenous expression in *E. coli*, we examined the oligomeric states of CLPB variants purified from HEK-

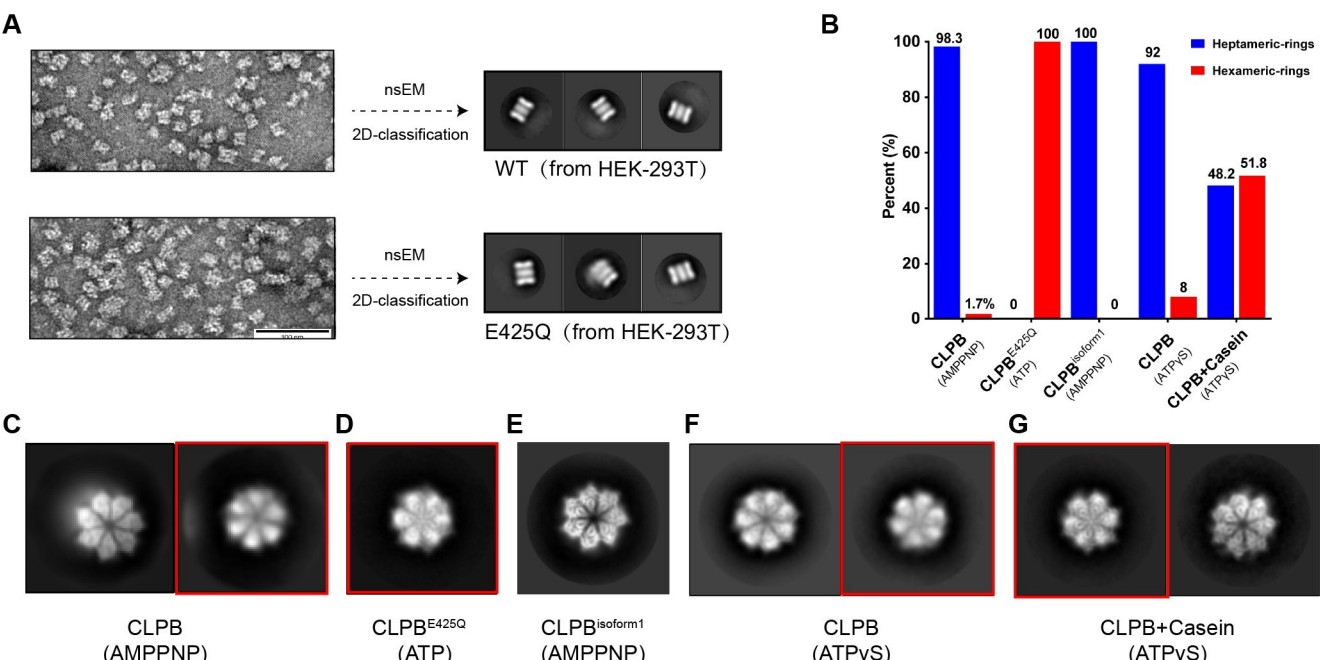

**Fig 2. The CLPB double-heptamers transform into double-hexamers upon substrate binding.** (**A**) CLPB complexes purified from HEK293 cells form double-oligomeric state. Representative nsEM images (left) and 2D classification averages of nsEM particles (right) of CLPB and CLPB[E425Q] from HEK-293T cells. (**B**) The proportion of the hexameric and heptameric top views in the CLPB-AMPPNP, CLPB[E425Q]-ATP, CLPB[isoform1]-AMPPNP, CLPB-ATPγS, and CLPB+Casein-ATPγS datasets (S2 Data). (**C–G**) Representative top views of 2D classification averages of CLPB-AMPPNP (C), CLPB[E425Q]-ATP (D), CLPB[isoform1]-AMPPNP (E), CLPB-ATPγS (F), and CLPB+Casein-ATPγS (G) datasets. The top views with hexameric feature are indicated by red boxes. nsEM, negative staining electron microscopy.

293T cells. The nsEM analysis indicates that both the WT and mutant CLPB complexes derived from human cells display a similar higher-order organization as the *E. coli* ones (Fig 2A). Next, to rule out the possible splicing isoform specific effect on the oligomeric state, we also examined the isoform-1 of CLPB (UNIPROT: Q9H078-1) purified from *E. coli* cells (S2E and S2F Fig) using both nsEM and cryo-EM. This isoform-1 is 30 residues longer in the ANK domain than the isoform-2. Our results show that CLPB[isoform1] complexes are again double-heptamers with an open central channel in the NBD ring (Figs 2E and S5C).

In short, our results and two recent independent studies [20,21] demonstrate that mitochondrial CLPB, unlike other typical AAA+ proteins, adopts a unique higher-order structure. The CLPB-specific ANK domain mediates the attachment of 2 heptamers/hexamers through a head-to-head dimerization. Both the double-heptamer and double-hexamer are highly dynamic. As also observed in another structural study of CLPB [20], several factors contribute to the conformational flexibility of the full-length complexes. The first is the hexamer/heptamer interface. Although the ANK domains from the 2 heptamer/hexamers interact with each other, they do not have lateral interactions within a single heptamer/hexamer. Thus, the interface between 2 heptamer/hexamers is rather flexible. The second is the flexibility within each heptamers/hexamers because the AAA + domain rings are known to adopt unsymmetrical and dynamic conformations.

## Substrate binding induces the transition from double-heptamer to double-hexamer

A major difference between the WT and E425Q CLPB structures is the presence of a peptide substrate in the central pore, which brings close CLPB subunits to form a more compact

structure. As shown in the 2D classification results, most of the top-view classes of the WT CLPB particles are heptameric, and only a tiny top-view class, 1.7% of all top-view particles, display a hexameric feature (Figs 2B, 2C, and S5A and S2 Data). In contrast, all the top-view classes of the CLPB$^{E425Q}$ particles are hexameric exclusively (Figs 2B–2D and S5B). This suggests that the binding of substrate may induce a transition from tetradecamers to dodecamers.

To test this hypothesis, we incubated the WT CLPB complexes with a model substrate casein [8] in the presence of excessive ATPγS, which has been shown to best promote the binding of casein to Hsp104 [7,31] and CLPB [20]. As a control, WT CLPB complexes were also treated with ATPγS alone. Cryo-EM 2D classification was employed to analyze their oligomeric states. For the CLPB-ATPγS dataset, only a small fraction (8.0% of all top-view particles) shows a hexameric ring (Fig 2B and 2F), indicating that the majority of particles retain the form of double-heptamer. In sharp contrast, in the presence of casein and ATPγS, the percentage of hexameric top-view particles has increased to 51.8% (Fig 2B and 2G). Consistently, Cupo and colleagues recently also found that a mixture of CLPB double-hexamers and double-heptamer in the present of casein and ATPγS. These results indicate that substrate binding is likely the most important factor that drives the formation of double-hexamers from double-heptamer.

## The ANK domain is essential for the assembly and function of CLPB complexes

To validate the role of the ANK domain in organizing the higher-order structure, we created an NBD-only variant by truncating the N-terminal ANK domain. Gel filtration and nsEM reveal that CLPB-NBD proteins still form oligomers (S9 Fig), but the size is much smaller than the double-heptamer or double-hexamer. Thus, these results and a similar study [20] experimentally proved a role of the ANK domain in assembling the higher-order CLPB complex. Next, we examined whether these CLPB-NBD oligomers have ATPase and disaggregase activities. Our results show that the disaggregase activity critically depends on the integrity of both the ANK and NBD domains. The introduction of E425Q mutation (CLPB$^{E425Q}$ or CLPB-NBD$^{E425Q}$) and the deletion of ANK domain (CLPB-NBD) both resulted in a complete loss of the disaggregase activity. This is consistent with a previous study [8], suggesting an essential role of the ANK domain in the disaggregase function of CLPB. However, in contrast to the same study, our data show that CLPB-NBD is active in hydrolyzing ATP, and the activity is even higher than the WT CLPB (S6 Fig and S3 Data). This discrepancy is due to the fact that CLPB-NBD in that study only formed dimer-like species under their pH 8.0 condition, whereas we used an acidic condition (pH 6.8) that is more relevant to that of the mitochondrial IMS.

For functional analysis of the ANK domain, we determined the crystal structure of the ANK domain at 2.1 Å resolution (S2 Table). The ANK domain of CLPB contains 4 AMs and a unique insertion (consisting of 2 short helices) between AM2 and AM3 (Fig 3A, right panel). The isoform 1 of CLPB differs from the isoform 2 exactly in this region, with 30 residues more in this insertion sequence (Fig 3A, left panel). Since the ANK domain is important for both the assembly and function of CLPB, we wonder whether this insertion is functionally relevant. From the density map of the double-hexamer, the substrate density extends from the NBD ring to the layers of ANK domains, and some extra density in the center of the ANK layers was observed (S7A Fig). Fitting of the crystal structure of the ANK domain reveals that the insertion exactly locates in the inner surface of the ANK layers. Interestingly, the insertion contains 2 patches of hydrophobic residues (Fig 3B), implying a potential role in substrate binding. Therefore, we constructed another CLPB mutant with the insertion completely deleted

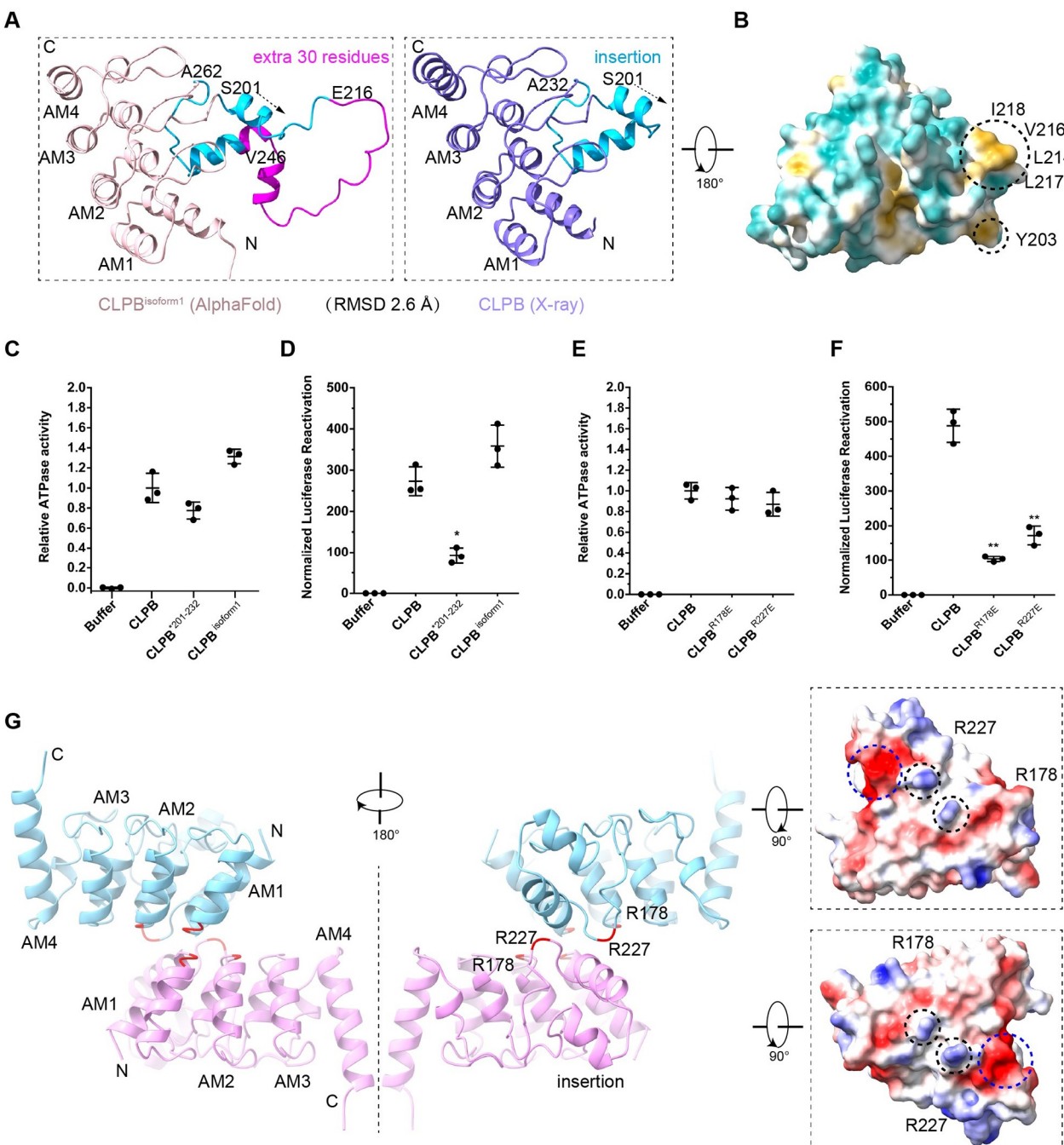

**Fig 3. The ANK domain is essential for the disaggregase activity of CLPB.** (**A**) X-ray crystallography structure and AlphaFold predicted model of the ANK domain. The RMSD between these 2 structures is 2.6 Å. (**B**) The hydrophobic surfaces of the ANK domain. Two hydrophobic patches were found in the unique insertion. (**C**) The ATPase assay of CLPB, CLPB$^{\Delta 201-232}$, and CLPB$^{isoform1}$. ATPase activity was compared to CLPB ($N = 3$, individual data points shown as dots, bars show mean ± SD) (S4 Data). (**D**) The disaggregase activity assay of CLPB, CLPB$^{\Delta 201-232}$, and CLPB$^{isoform1}$. Disaggregase activity was compared to CLPB ($N = 3$, individual data points shown as dots, bars show mean ± SD, $^*p < 0.05$) (S4 Data). (**E**) ATPase assay of CLPB, CLPB$^{R178E}$, and CLPB$^{R227E}$ (S4 Data). (**F**) Disaggregase activity assay of CLPB, CLPB$^{R178E}$, and CLPB$^{R227E}$. Results show that the disaggregase activities of CLPB$^{R178E}$ and CLPB$^{R227E}$ are reduced by 70%–80% ($N = 3$, individual data points shown as dots, bars show mean ± SD, $^{**}p < 0.01$) (S4 Data). (**G**) The ANK domain dimer interface. One is the AM1-AM2 and the other is the second helix of the insertion. Two positively charged residues (R178 and R227) are labeled as black dashed circles on the ANK surface electrostatic potential.

(CLPB$^{\Delta 201-232}$) and compared this mutant with both isoforms 1 and 2 in their ATPase and disaggregase activities. The CLPB$^{\Delta 201-232}$ mutant showed the same oligomeric state as the isoforms 1 and 2 (S7C–S7E Fig). The 2 isoforms have comparable activities in these 2 molecular functions (Fig 3C and S4 Data). Although the deletion in CLPB$^{\Delta 201-232}$ had no effect on the ATPase activity, it reduced disaggregase activity sharply by more than 50% (Fig 3C and 3D and S4 Data). Therefore, this insertion is likely involved in the substrate recognition and may also contribute to the translocation of the substrate to the C-terminal NBD ring. In agreement with our data, Cupo and colleagues also found that the deletion of this loop (Skd3$^{\Delta L}$), as well as the deletion of the first 2 AMs or the last 2 AMs, all resulted in more than 50% of reduction in the disaggregase activity [20].

Next, we analyzed the dimer interface of the ANK domain by docking the ANK models into the density maps of the double-heptamer and double-hexamer (S8A and S8B Fig). The ANK dimers in the 2 higher-order structures are not identical but generally similar. For the ANK dimer in the double-hexamer, 2 regions of the ANK domain contribute to the dimerization, the AM1/2 motifs and the insertion (Fig 3G). One interface is relatively extensive and formed by 4 connecting loops between the first and second helices of the AM1 or AM2 from the 2 opposite ANK domains. The other interface is mediated by the second helix (and its downstream flanking sequence) of the insertion, and 2 arginine residues, R178 and R227, appear to be important in this interface. Based on the electrostatic surface potential, these 2 residues of 1 ANK domain point to a highly negatively charged surface patch (E140, E221, D222, and D223) of the opposite ANK domain (Fig 3G). In fact, both R178 and R227 are highly conserved among the metazoan species [8]. Therefore, we performed mutagenesis (R178E and R227E) to test whether they could disrupt the ANK dimerization. Unexpectedly, the 2 CLPB mutants still maintain a higher-order assembly, but their disaggregase activities are severely impaired by 70% to 80% (Figs 3E, 3F, and S8C–S8E and S4 Data). These results further emphasize a role of the ANK domain in the disaggregase function of CLPB.

## CLPB-NBD assembles into polypeptide-engaged helical structures

Due to the large inter- and intra-hexamer/heptamer flexibility, the structures of full-length CLPB complexes were not solved in atomic resolution. CLPB-NBD was thus used as a surrogate for high-resolution structural determination. Two mutant versions of CLPB (NBD and NBD$^{E425Q}$) were analyzed by cryo-EM (S9A–S9F Fig), which again confirmed that E425Q mutation resulted in co-purification of an endogenous peptide in the central channel. Also similar to the full-length complexes, the top-view classes of the NBD$^{E425Q}$ particles are exclusively hexameric, whereas both hexameric and heptameric arrangements were observed for the NBD$^{WT}$ particles (S9F Fig). Subsequently, we focused on the NBD$^{E425Q}$ dataset for high-resolution refinement. After several rounds of 3D classification, 3 different oligomeric arrangements, hexameric, heptameric, and nonameric, were identified. The nonamer appeared to be more stable and could be resolved at an overall resolution of 3.7 Å (S10 Fig), allowing the atomic modeling of the NBD of CLPB. In the nucleotide-binding pockets of the nonamer, a total of 8 ATP molecules could be modeled (S11 Fig). The conserved functional motifs on the ATPase domain are well resolved, including the Walker A motif (351-GSSGIGKT-358), sensor-1 (464-TSN-466), and sensor-2 (588-GAR-590) (Fig 5E). The conserved residues I317, I318, and F541 form a hydrophobic pocket to stabilize the adenine ring of ATP, while T358 stabilizes the β- and γ-phosphates through a Mg$^{2+}$ ion (Fig 5E). The Arginine Finger (R531) from the adjacent protomer points to the γ-phosphate of ATP. In general, these structures show that the NBD of CLPB is a typical AAA+ ATPase module, underscoring a conserved mechanism of ATP-powered substrate unfolding and translocation [32].

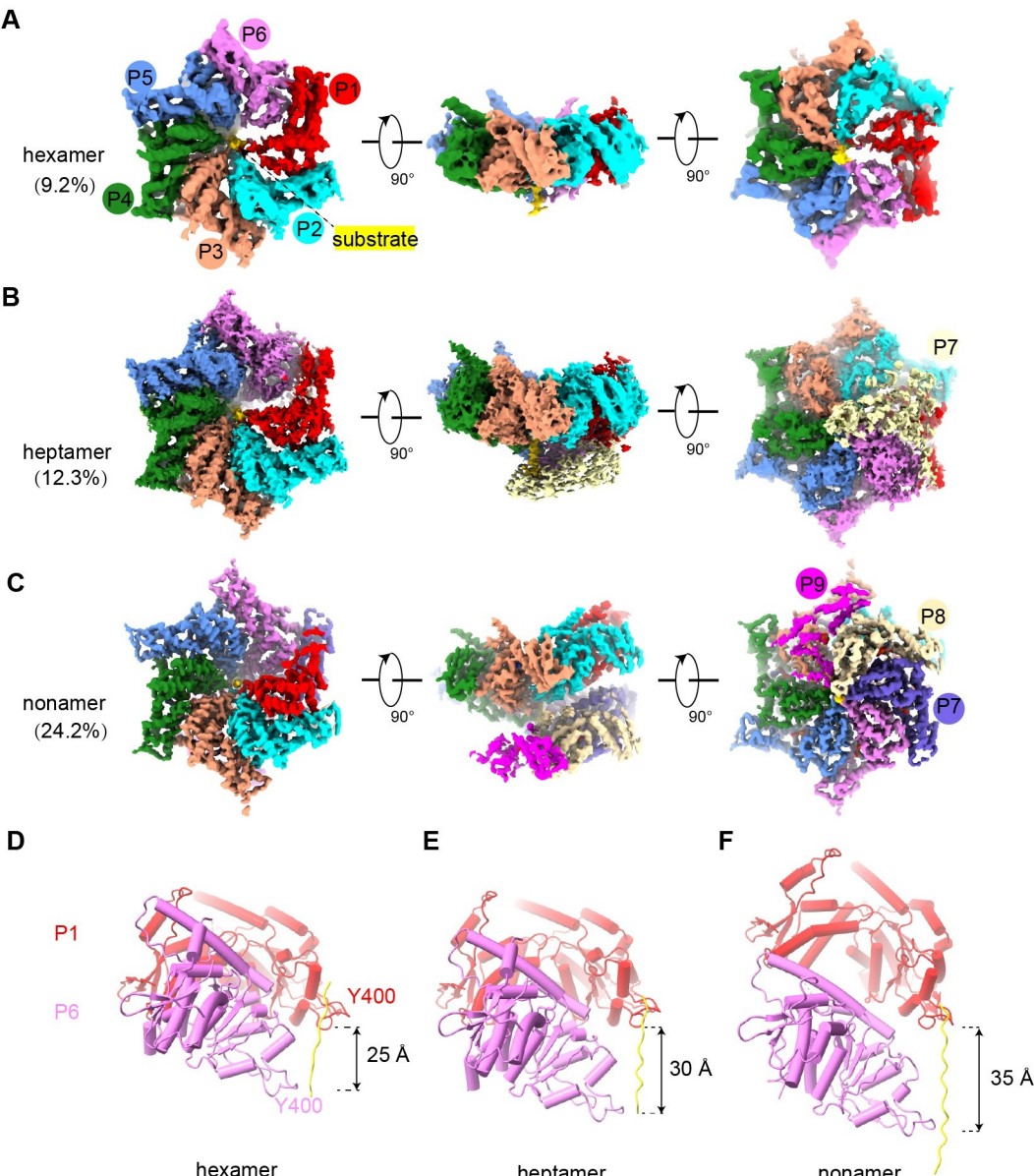

**Fig 4. Cryo-EM characterization of the NBD helical structures of CLPB.** (A–C) Density maps of the NBD hexamer (A), heptamer (B), and nonamer (C), respectively. Protomers are painted in different colors. (D–F) Distances along the central substrate between the PL residue Y400 of P1 and P6 in the hexamer (D), heptamer (E), and nonamer (F). The axial rise of P6 relatively to P1 are labeled. NBD, nucleotide-binding domain; PL, pore-loop.

The central channels of all 3 states are occupied by a peptide substrate, which forms successive interactions with the helically arranged CLPB protomers (Fig 4A–4C). In the nonamer, 9 protomers form a 1.5-turn helix around the central substrate. Besides the protomer number, the 3 oligomers also show obvious structural differences. While the twist angle between neighboring subunits is roughly 60° in all 3 forms, the axial rise within each oligomer is not constant and varies in the hexamer and heptamer largely. The axial displacements between P1 and P6 are roughly 25 Å, 30 Å, and 35 Å for the hexamer, heptamer, and nonamer, respectively (Fig 4D–4F). This is expected, as more axial space is required to fit in the seventh and more

subunits onto the hexamer. In the full-length complexes, the ANK domain provides a steric hindrance to end the helical extension.

Overall, these data show that the ANK domain is essential in determining the oligomeric state of CLPB complex and also explain why our CLPB-NBD still retains ATPase activity. Although this nonameric helical structure does not exist in a physiological context, it serves as a high-resolution model to demonstrate that CLPB-NBD shares many characteristics of typical AAA+ proteins.

## The substrate interacts with conserved pore-loops of the helically arranged NBDs

The density of the polypeptide backbone is well resolved in the nonamer and was modeled as a 17-residue poly-alanine fragment (Fig 5A and 5B). The peptide spans about 53 Å length in the central channel of the nonamer and displays successive interactions with the PL-I and PL-II from NBD protomers (Fig 5B). For each protomer, a canonical PL-I (399-GYVG-402) binds to the peptide through 2 hydrogen bonds formed between the main-chain atoms (Fig 5C). The first is between the carbonyl oxygen of the substrate residue N and the main-chain nitrogen of V401, and the other between the carbonyl oxygen of G399 and the main-chain nitrogen of the substrate residue N+2 (Fig 5C). This pattern of interactions is nearly identical for all protomers, except that the hydrogen bond distances vary between 2.5 to 4.0 Å. Thus, the axial step size of PLs is exactly 2 residues. Of note, the main-chain atoms of Y400 do not participate in sequence-independent interaction with the substrate backbone. Therefore, the essential role of Y400 in the disaggregase activity should arise from its aromatic side-chain, which could interact with various side-chains of a threaded substrate. Moreover, the substrate could be modeled in both directions (N to C or C to N) in the density map, and both configurations could satisfy this repeated pattern of interactions (Fig 5C and 5D). In contrast to the PL-I, the PL-II, consisting of E386, R387, and H388, is relatively away from the substrate backbone, except that the side-chain of H388 is within 4-Å distance with the β-carbon atom of the substrate (Fig 5B). Thus, it is likely that the primary role of PL-II during substrate processing is to interact with different side-chains of the substrate through its polar residues.

In short, our results and a recent study [20] indicate that the NBD ring of CLPB is a typical unfoldase core, and the PLs of CLPB functions to grip and move the substrate within the central channel in a conserved manner as classic AAA+ unfoldases/disaggregases.

## Mitochondrial interactome analysis of the ANK domain

Previous data showed that *CLPB* knockout cells exhibited decreased solubility for many proteins in the IM and IMS of mitochondria, including HAX1, TOMM22/70, TIMM22/23, HTRA2, PHB1/2, OPA1, STOML2, and SLC25 family proteins [8]. Given the essentiality of the ANK domain in the disaggregase activity of CLPB, we performed a mass spectrometry (MS)-based interactome analysis of the ANK domain. A plasmid harboring the MTS and ANK domain sequences was expressed in HEK-293T cells, and mitochondria were then isolated, lysed, and dissolved in 1% detergent. The ANK domain and its binders were purified through a C-terminal Strep tag, and subject to MS analysis (Fig 6A). A relatively stringent criterion (fold-change > 4 and *p*-value < 0.01) were used for enrichment analysis (3 biological replicates). With DAVID Bioinformatics Resources [33], we restricted our analysis on mitochondrial proteins detected in the samples. Compared with the control sample, 934 mitochondrial proteins were significantly enriched in the ANK sample (Fig 6B and S3 Table and S5 Data), and most of them are located in the IM and matrix (60.3%). Although CLPB is located in the IMS, it may also function to promote the solubility of certain membrane proteins with IMS-

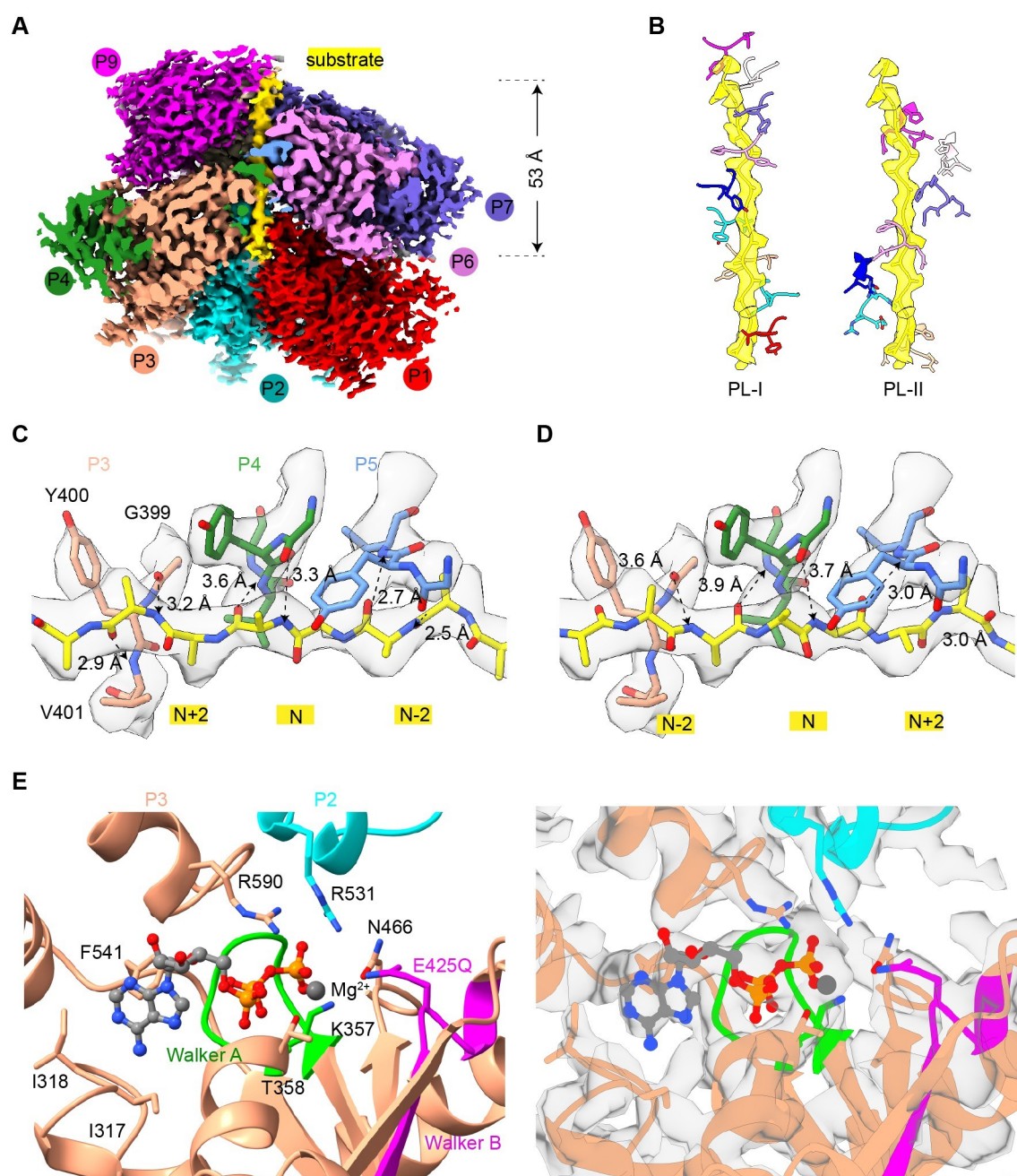

**Fig 5. Structure of the NBD nonamer in processing a peptide substrate.** (**A**) Density map of the NBD nonamer in the substrate-processing state. Nine protomers are indicated as P1 to P9 and painted in different colors. The central substrate is colored yellow. (**B**) The spiral configuration of the PLs of CLPB protomers around the substrate in the central channel. PL-Is and PL-IIs are shown in the left and right panels, respectively. While the conserved PL-Is (G399-Y400-V401-G402) directly interact with the substrate, the PL-IIs (E386-R387-H388) situate in slightly larger distances from the substrate. (**C, D**) Magnified view of the interactions between the PL-I of P3, P4, and P5 and the backbone of substrate. The substrate could be modeled in both directions, N to C or C to N. The potential hydrogen bonds are indicated by dashed lines and the distances are labeled. (**E**) Magnified view of the conserved ATP-binding pocket of CLPB. Functionally important residues of Walker A motif (K357, T358), Walker B motif (E425), sensor-1 (N466), sensor-2 (R590), arginine finger (R531), and the conserved residue I317, I318. Atomic model and density map are shown in the left and right panels, respectively. NBD, nucleotide-binding domain; PL, pore-loop.

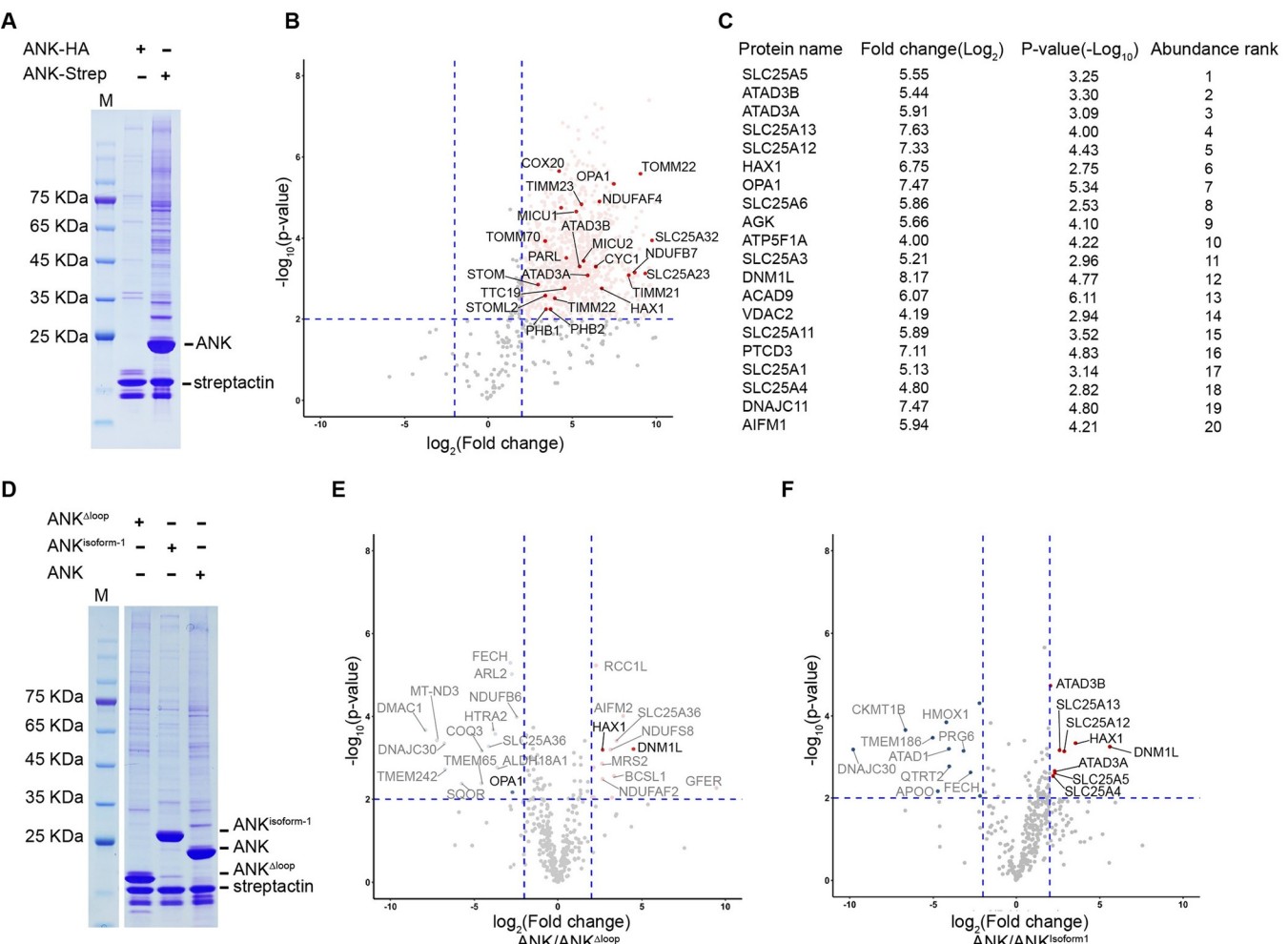

**Fig 6. The mitochondrial interactome of the ANK domain.** (**A**) HEK-293F cells expressing the MTS-ANK-HA or MTS-ANK-Strep constructs were subjected to pulldown experiments with Strep resins. Precipitates were analyzed by SDS-PAGE, Coomassie blue staining, and MS. (**B**) Volcano plot showing the mitochondrial proteins co-precipitated with MTS-ANK-Strep. A total of 935 proteins that were enriched in the MTS-ANK-Strep are labeled in light red. The potential substrates, previously reported to be most affected by *CLPB*-knockout, are highlighted in red. A relatively stringent criterion (fold-change > 4 and *p*-value < 0.01) were used for enrichment analysis (3 biological replicates), indicating with blue dashed lines (S5 Data). (**C**) Top 20 proteins in OM, IMS, and IM are listed based on the ranking of protein abundances (S4 Table). (**D**) HEK-293F cells expressing the MTS-ANK-Strep, MTS-ANK^Δloop-Strep, and MTS-ANK^isoform1-Strep constructs were subjected to pulldown experiments with Strep resins. Precipitates were analyzed by SDS-PAGE, Coomassie blue staining, and MS. (**E, F**) Volcano plot showing the fold change of the OM, IMS, and IM mitochondrial proteins in MTS-ANK-Strep compared to MTS-ANK^Δloop-Strep (E) or MTS-ANK^isoform1-Strep (F). Proteins in the top 20 list (C) were heighted in red (S6 and S7 Data). IMS, intermembrane space; MS, mass spectrometry; MTS, mitochondrial targeting signal.

exposed domains. Therefore, we focused on proteins from the OM, IMS, and IM. As a result, 381 such proteins were selected for further analysis, including all of those potential substrates previously reported to be most affected by *CLPB*-knockout [8]. If these proteins were ranked by their abundances, 3 IMS proteins stood out in the top 20 list, including HAX1, OPA1, and AGK (Fig 6C and S4 Table). Consistently, both HAX1 and OPA1 have been experimentally validated to interact with CLPB [18,34]. In general, the ANK interactome identified in this study agrees well with previous proteomic data based on full-length CLPB [18,34], indicating that the ANK domain has a direct role in substrate binding.

We also showed that the unique insertion between AM2 and AM3 in the ANK domain is important in the disaggregase activity in vitro (Fig 3). Therefore, 2 ANK variants were next

used for interactome analysis. The MTS-ANK$^{\Delta201-232}$ and MTS-ANK$^{isoform1}$ were expressed in HEK-293T cells and their interactome were similarly analyzed by MS (Fig 6D). Among the analyzed mitochondrial OM, IMS, and IM proteins, 8 proteins were more enriched in the ANK$^{isoform2}$ sample, including ATAD3B/ATAD3A, SLC25A4/5/12/13, HAX1, and DNM1L (Fig 6F and S6 Data). All of them are indeed among the most abundant top 20 list. In contrast, a larger number of proteins (14) were much more enriched in the ANK$^{isoform2}$ sample than the ANK$^{\Delta201-232}$ sample, again including HAX1 and DNM1L (Fig 6E and S7 Data). Highly consistent with our findings, a recent study reported that HAX1 preferentially binds to the isoform 2 of CLPB in vivo, rather than the isoform 1 [34].

These interactome analyses support a direct role of the ANK domain in substrate binding, and together with previous studies [8,18,34] confirmed that HAX1 is one of the natural substrates of CLPB. The differential enrichment between the isoform1 and 2 further suggests a possible role of the ANK insertion in substrate selection, probably through altered binding affinity.

## Discussion

In the present work, we characterized the structures of CLPB in both the apo- and substrate-bound states. Unexpectedly, we found that apo CLPB assembles into higher-order structures, in the form of a double-heptamer through inter-molecular interactions mediated by N-terminal ANK domains (Fig 1). This high-order organization is consistent with recent structural studies of purified CLPB [20,21] and a previous native gel analysis of endogenous CLPB proteins [11]. We further demonstrated that the double-heptamers could be efficiently converted to double-hexamers upon the addition of a model substrate and ATPγS (Fig 2). It is not clear whether the double-heptamer is a physiological state of CLPB in vivo. But several lines of evidence appear to suggest it is not a pure in vitro artifact. First, the heptamer form is the predominant species of apo CLPB (Fig 2), indicating that it is an intrinsic property of the CLPB ATPase module. Although AAA+ proteins are generally considered to be hexamers, an increasing number of AAA+ proteins were recently discovered to take heptamer as a primary oligomeric state [23,24,35,36], including a mitochondrial IM-bound AAA+ protein Bcs1 [23,24]. Second, the substrate-induced change on the oligomeric state has also been observed for other AAA+ members, such as DNA helicase AAV2 Rep68 [37], RuvB [38], and archaeal MCM [39]. Third, the ATPase activity of full-length CLPB (double-hexamer) is lower than CLPB-NBD alone (S6 Fig). In fact, roughly two-thirds of CLPB-NBD particles exhibit a hexameric arrangement. This indicates that the heptamer is not optimized for efficient ATP-hydrolysis. Thus, as a substrate-free state, the heptameric form could avoid necessary ATP consumption in the IMS. Notably, the buffer we used to prepare CLPB complexes was at pH 6.8, which closely resembles that of the IMS [40]. Therefore, we propose that if the heptameric form is not an in vitro artifact, it could be a resting state of CLPB in mitochondria that could respond to the ATP concentration and the availability of substrates to change its oligomeric states. This interesting hypothesis merits further investigation.

With these said, the core of substrate-engaged CLPB is still hexameric as shown in the structures of CLPB$^{E425Q}$ and NBD (Figs 1 and 4) [20]. In the high-resolution structures of the NBD$^{E425Q}$, the PL-Is of the protomers form a spiral staircase around the substrate in a two-residue step-size (Fig 5C and 5D). These PL-Is interact with the substrate in a sequence-independent manner and capable of accommodating polypeptide in both the N-C and C-N directions. This implies that CLPB could potentially thread the substrate in both directions, similar as bacterial ClpX and ClpA [41]. These data indicate that the mechanisms of CLPB in substrate

threading and translocation are highly similar to those cytosolic AAA+ unfoldases/disaggregase, such as Cdc48/p97 and Hsp104 [7,25,29,30].

In the double-hexamer, both hexamers are in active conformation with the substrate being threaded through the central pores of their ATPase rings. This raises an interesting question: Whether or not the 2 hexamers work in a synergistic manner, since a CLPB hexamer already contains all essential structural features of a unfoldase/disaggregase. A conventional model is that the 2 hexamers work independently, and upon engagement with a protein aggregate the 2 hexamers do not need to maintain stable association for all 6 pairs of ANK domains. However, as seen in the structure of the double-hexamer (Fig 1D), it is also possible that the axial movement of a CLPB subunit in 1 hexamer during the ATP-hydrolysis cycle could potentially affect another CLPB subunit in the other hexamer through their tightly associated ANK domains. Unfortunately, our attempt to obtain ANK mutants to separate its roles in structural organization and disaggregase activity has failed (Fig 3E and 3F). It is possible that a complete disaggregase cycle of CLPB may involve dynamic remodeling of the dimer interface or even disassociation/re-association of hexamers [20]. Moreover, this "head to head" organization reminds us of the recent findings on the double-hexamer form of p97 [42–45], although p97 utilizes a "back to back" interface through 2 C-terminal ATPase rings. These 2 examples underline a possibility that AAA+ proteins could adopt different higher-order assemblies to serve different purposes.

We also determined the crystal structure of the ANK domain and analyzed its functional significance in the disaggregase function. Our data showed that the ANK domain is indispensable for the disaggregase activity of CLPB, and a deletion of the unique insertion in the ANK domain resulted in more than 50% reduction on the disaggregase activity (Fig 3D). Furthermore, 2 single mutations of the ANK domain in the ANK dimer interface, including R178E and R227E, both greatly impaired the disaggregase activity by 70% to 80% (Fig 3F). R227 is exactly located to the insertion sequence. These observations suggest that this unique insertion is functionally important. Since this insertion is at the innermost position of the ANK ring and distant from the PLs of the ATPase modules, the ANK domain may function to recognize and pass the substrate to the distal ATPase ring. This hypothesis was further supported by our interactome analysis of the separate ANK domain. A large number of the IM and IMS proteins were greatly enriched in the sample affinity-purified through tagged ANK, including all of those previously found to be prone to aggregation in $\Delta CLPB$ cells [8]. Many of these candidates have also been reported to interact with CLPB in previous large-scale proteomics studies [46,47] and 2 of them have been experimentally validated, such as HAX1 and OPA1 [18,34]. Of note, several mitochondrial proteins involved in cristae remodeling are among the most enriched group, including OPA1, DNM1L, and IMMT. This appears to collaborate with the previous finding that CLPB loss resulted in altered cristae structure in mitochondria [18]. We also tested the $ANK^{\Delta 201-232}$ and $ANK^{isoform1}$ in interactome analysis. Compared with the $ANK^{isoform2}$, a relative extensive change on the enrichment profile was observed for the $ANK^{\Delta 201-232}$ sample. In contrast, comparison between isoform 1 and isoform 2 reveals that many potential substrates of CLPB are significantly more enriched in the sample of $ANK^{isoform2}$, including HAX1 and DNM1L (Fig 6E and 6F). These results further pinpoint a role of the ANK-insertion in the substrate recruitment and suggest another level of regulation on CLPB function through alternative splicing.

The structures of CLPB also allow us to map disease-related mutations [8,10,13–16,48,49] on the atomic model to understand their possible effect on the function of CLPB. Mutations associated with 3-MGA could be categorized into 3 classes. The first class of mutations lie at the interface between 2 adjacent CLPB protomers, including R378G, R445Q, Y587C, R598C, and E609K (Fig 7A). These mutations likely perturb the inter-protomer communication to

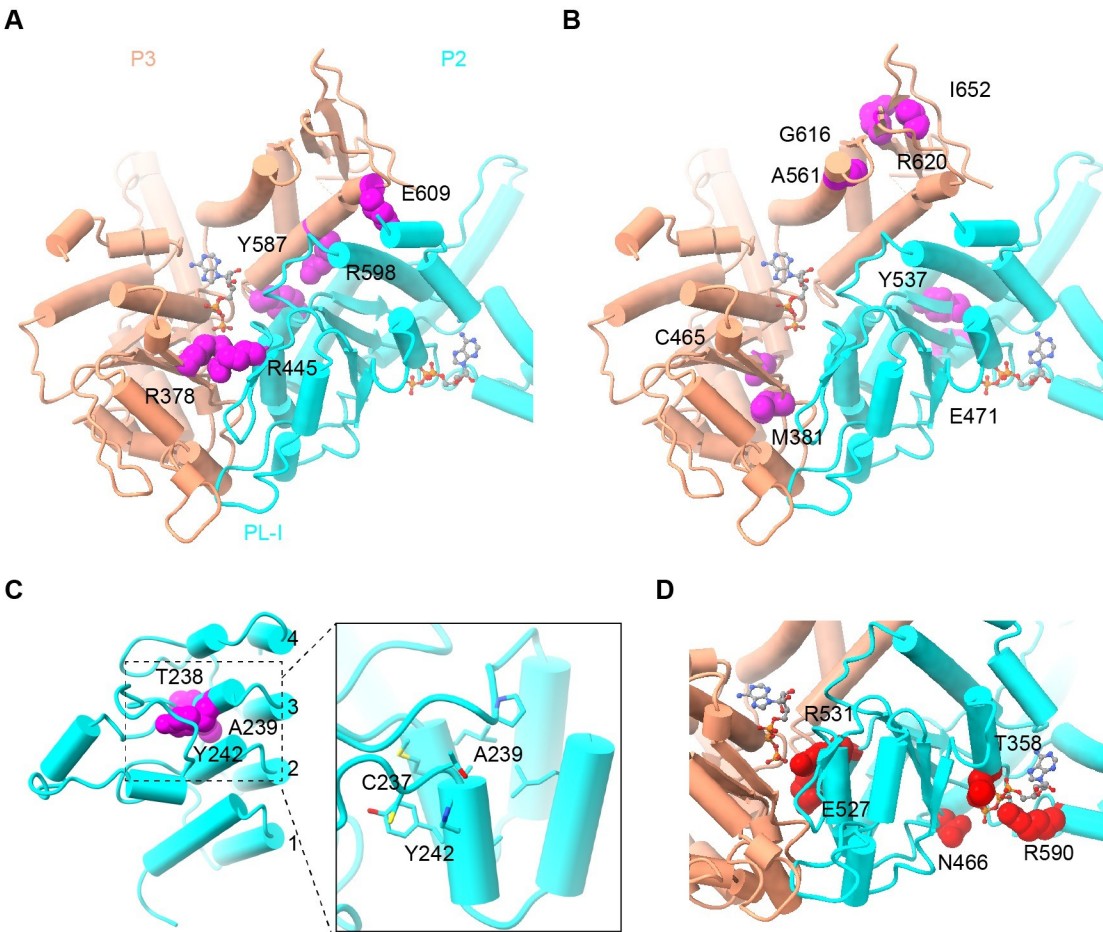

**Fig 7. The disease-related mutations of CLPB in 3-MGA and SCN.** (**A**) 3-MGA-related mutations in the interface of adjacent protomers. (**B**) 3-MGA-related mutations within the large and small subdomain of the ATPase domain. (**C**) 3-MGA-related mutations in the ANK domain. (**D**) SCN-related mutations in the ATP-binding pocket. Residues on the positions of these mutations are highlighted in magenta or red sphere models. ATP molecule is highlighted in stick models. SCN, severe congenital neutropenia; 3-MGA, 3-methylglutaconic aciduria.

impair CLPB function. The second class consists of M381I, C456R, E471K, and Y537C in the large subdomain, and A561V, G616V, R620P, and I652N in the small subdomain (Fig 7B). These mutations are away from the inter-protomer or inter-domain interfaces, and they may destabilize the domain structure of CLPB (Fig 7B). The third class of mutations are in the AM3 of the ANK domain. Besides 2 nonsense mutations (R250* and K321*), 3 mutations (T238M, A239T, and Y242C) are on the first helix of the AM3 (Fig 7C). This region contains a characteristic tetrapeptide (-TPLH-) motif (238-TALH-241 in CLPB) and is highly conserved among AMs [50]. Therefore, these mutations might destabilize the structure of the ANK domain or lead to misfolding of the AMs. Consistently, it was shown that T238M did not affect the ATPase activity, but largely inhibited the disaggregase activity [8]. As for mutations related to SCN, they are exclusively located close to the active center of the NBD (Fig 7D). They are distributed in sensor-1 (N466K), sensor-2 (R590C), AF (R531Q/G), and Walker A motif (T358K). It is apparent that they may affect the binding and hydrolysis of ATP (Fig 5E). Indeed, CLPB with these mutations were defective in both the ATPase and disaggregase activities in vitro [17].

## Methods

### Protein expression and purification

The coding sequence (isoform 2) of human *CLPB*, amplified from a homemade cDNA library of HEK-293T cells, was cloned into pET28a vector with an N-terminal His×6-SUMO tag followed by a TEV protease cleavage sequence and expressed in *E. coli* Transetta (DE3) cells (TransGen Biotech). CLPB protein variants were purified as reported previously [9], with some modifications. Briefly, the culture was induced with 1 mM IPTG at $OD_{600}$ of 0.6 to 0.8 at 20˚C overnight. Cells were harvested and resuspended in lysis buffer I (50 mM Tris-HCl (pH 8.0), 500 mM NaCl, 5% glycerol, 1% Triton X-100, and 1 mM PMSF) and lysed by sonication. Lysates were cleared by centrifugation in a JA-25.50 rotor (Beckman) for 30 min at 20,000 r.p.m., and the supernatants were precipitated with ammonium sulfate at 35% saturation and incubated for 10 min at room temperature (RT). The cloudy lysates were centrifuged for 30 min at 15,000 r.p.m., and protein pellets were dissolved with Buffer B (50 mM Tris-HCl (pH 8.5), 5 mM DTT). The samples were then purified using a Mono Q column (GE Healthcare), and the bound proteins were eluted with a linear gradient to 60% Buffer C (50 mM Tris-HCl (pH 8.5), 1 M NaCl). To remove the His×6-SUMO tag, TEV protease was added into the eluates and incubated at RT for 2 h. The tags and contaminations were removed by size-exclusion chromatography (Superose 6 Increase 10/300 GL, GE Healthcare) using the elution buffer (20 mM HEPES-KOH (pH 6.8), 300 mM KCl, 5 mM $MgCl_2$). Peak fractions were pooled, concentrated, and incubated with 5 mM nucleotide (AMPPNP for CLPB, ATP for $CLPB^{E425Q}$) at RT for 2 h. Approximately 1 mM BS3 was added to improve the sample stability, and the sample was further subjected to size-exclusion chromatography (Superose 6 Increase 10/300 GL, GE Healthcare). Peak fractions were first examined by nsEM, pooled, concentrated, and mixed with 1 mM nucleotide for cryo-EM grids preparation. $CLPB^{E425Q}$ and $CLPB^{Y400A}$ were generated by site-specific mutagenesis, and the samples for cryo-EM grids preparation were similarly purified.

For preparation of NBD and $NBD^{E425Q}$ proteins, harvested cells were resuspended in lysis buffer II (50 mM Tris-HCl (pH 7.4), 300 mM NaCl, 5% glycerol, 1% Triton X-100, 1 mM PMSF, and 20 mM imidazole) and lysed by sonication. The supernatants were incubated with Ni–NTA agarose beads (GE Healthcare) at 4˚C for 2 h. After washing with lysis buffer for 5 times, proteins were eluted with elution buffer (50 mM Tris-HCl (pH 7.4), 300 mM NaCl, 5 mM $MgCl_2$, and 500 mM imidazole). TEV protease was added into the eluates to remove His×6-SUMO tags and incubated at RT for 2 h. The proteins were loaded onto a size-exclusion column (Superose 6 Increase 10/300 GL, GE Healthcare), which was pre-equilibrated with 20 mM HEPES-KOH (pH 6.8), 300 mM KCl, 5 mM $MgCl_2$. Peak fractions were pooled, concentrated, and incubated with 5 mM nucleotide (AMPPNP for NBD, ATP for $NBD^{E425Q}$) at RT for 2 h, and the sample was then loaded onto a size-exclusion column. Peak fractions were examined by nsEM, pooled, concentrated, and mixed with 1 mM nucleotide for cryo-EM grids preparation.

For purification of the ANK domain, cells were resuspended in lysis buffer III (50 mM Tris-HCl (pH 8.0), 500 mM NaCl, 5% glycerol, 1% Triton X-100, 1 mM PMSF, and 20 mM imidazole) and lysed by sonication. The supernatants were incubated with Ni–NTA agarose beads (GE Healthcare) at 4˚C for 2 h. The eluates were loaded onto a size-exclusion column (Superdex 75 10/300 GL, GE Healthcare) and eluted with 50 mM Tris-HCl (pH 8.0), 150 mM NaCl.

### ATPase activity measurement

Approximately 0.1 μm WT CLPB or CLPB variants were incubated with 1 mM ATP at 30˚C for 30 min in ATPase activity buffer (20 mM HEPES-KOH (pH 6.8), 150 mM KAOc, 5 mM

MgCl$_2$, 0.1% Tween-20). ATPase activity was measured using the malachite green phosphate assay [51]. For each protein, at least 2 biological replicates were measured with 3 independent replicates.

## Luciferase disaggregation assay

Disaggregase activity was measured as previously reported [8]. Briefly, 1 μm WT CLPB or CLPB variants were incubated with 50 nM freshly prepared firefly luciferase aggregates at 30˚C for 90 min in luciferase reactivation buffer (20 mM HEPES-KOH (pH 8.0), 150 mM KAOc, 10 mM KAOc, 10 mM DTT, 5 mM ATP). Recovered luminescence was monitored with the luciferase reporter assay kit. The disaggregase activity of CLPB in S1F Fig was detected using the kit from Beyotime Biotech (CAT# RG005), and the measurements in other figures were done with the kit from TransGen Biotech (CAT# FR101-01).

## Transient overexpression of CLPB variants

HEK-293 T/F cells were transfected with pCAG vector containing CLPB variants. Cells were transfected using PEI transfection reagent and harvested after 48 h.

## Western blots

Cells were washed with PBS and incubated with lysis buffer (50 mM Tris-HCl (pH 7.4), 150 mM NaCl, 1% Triton X-100, 0.1% SDS, and 1 mM PMSF) for 30 min on ice. α-Strep antibody (Huaxing Bio, CAT# HX1816) was used to detect the target protein.

## Mitochondrial isolation and preparation of the ANK pulldown samples

Mitochondria were isolated as previously described [8,52]. Briefly, cell pellets from 800 mL culture were resuspended in 40 mL mitochondria isolation buffer (20 mM HEPES-KOH (pH 7.6), 250 mM sucrose, and 2 mM EDTA) and homogenized with a Dounce homogenizer at 4˚C. Lysates were centrifuged at 1,300g for 5 min. The supernatant was collected and centrifuged at 13,000g for 15 min. And the pellet was resuspended with lysis buffer (20 mM Tris-HCl (pH 8.0), 150 mM NaCl, 1% Triton X-100, and 1 mM PMSF) and incubated for 30 min at 4˚C. Lysates were centrifuged at 20,000g for 30 min, and the supernatants were collected and incubated with Strep resin for 2 h at 4˚C. After extensive washing, SDS-loading buffer was added to the sample. The samples were separated by MOPS-PAGE on a gradient gel (4% to 20%).

## LC-MS/MS analysis

The samples were excised into several parts form protein gels, reduced with DTT, alkylated with iodoacetamide (IAA), and subsequently digested with trypsin. Peptides were analyzed by Thermo Orbitrap Exploris 480 mass spectrometer (Thermo Fisher Scientific) coupled with an Easy LC system (Thermo Fisher Scientific). Peptides mixtures were loaded onto C18 Trap Column in Buffer A (0.1% formic acid) and separated with a gradient Buffer B (0.1% formic acid, 80% ACN) (2 min 4% to 8% B; 37 min 8% to 25% B; 11 min 25% to 35% B; 7 min 35% to 95% B; 3 min 95% B) at a flow rate of 300 nL/min. Data were acquired in data-dependent mode with the following settings: MS1 60,000 resolution, 300 to 1,650 m/z of mass range; MS2 30% normalized collision energy, 15,000 resolution, 1.6 m/z of isolation window. Proteins were identified and quantified by Proteome Discoverer software 2.2 using default settings against a database of UNIPROT *Homo sapiens*. Methionine oxidation and N-terminal acetylation were

set as variable modifications. Protein-level and peptide-level false discovery rate (FDR) was set at 1%.

The differential enrichment analyses were based on 3 biological replicates (starting from expression plasmid transfection) and a stringent screening condition (*p*-value < 0.01 and Fold change > 4) was used. In the MTS-ANK-Strep samples, 1,115 mitochondrial proteins were detected, of which 934 proteins were highly enriched, compared with the MTS-HA sample (S5 Data). From these highly enriched proteins, those located in the OM, IMS, and IM were selected for further analysis (S4 Table and Fig 6C). The comparison between the MTS-ANK-Strep and MTS-ANK$^{\Delta201-232}$-Strep samples was focused on 374 mitochondria proteins with OM, IMS, and IM locations (S7 Data). The comparison between the MTS-ANK-Strep and MTS-ANK$^{isoform1}$-Strep samples was similarly done, and 332 OM, IMS, and IM proteins were used for analysis (S6 Data).

## Cryo-EM grids preparation

Cryo-grids were prepared with an FEI Vitrobot Mark IV with the chamber set at 6°C and 100% humidity. A total of 4 μL aliquots of the prepared samples described above were applied onto the glow-discharged holey-carbon grids (R1.2/1.3, Au, 300 mesh, Quantifoil) and blotted after a 30-s waiting time. The grids were then flash frozen using liquid ethane. The cryo-grids were screened using an FEI Talos Arctica microscope operated at an accelerating voltage of 200 kV, and the qualified ones were recovered for data collection. To address the issue of strong preferred orientation of the particles distribution in the cryo-grids for CLPB and CLPB$^{E425Q}$ samples, 0.5 mM CHAPSO was added to these 2 samples, as described previously [53].

## Cryo-EM data collection and image processing

For the sample of CLPB-AMPPNP, micrographs were collected using a 300-kV FEI Titan Krios microscope (Gatan GIF K3 camera) at a nominal magnification of 81,000× (calibrated pixel size of 1.07 Å). A total of 40 frames with a total exposure time of 3.2 s at a dose rate of 20 e$^-$/pixel/s were collected (defocus ranging from −1 to −1.4 μm). A total of 9,341 micrographs were collected automatically using the EPU software (Thermo Fisher Scientific). The micrographs were subjected to beam-induced motion correction using MotionCor2 [54], and contrast transfer function (CTF) parameters for each micrograph were determined by Gctf [55]. RELION (v3.1) was used to perform data processing [56]. After auto-picking, the particles were subjected to 2D classification and several rounds of preliminary 3D classifications. A total of 299,306 particles were selected and subjected to the next round of fine 3D classification (K = 4), and a final set of 231,212 particles were kept for 3D refinement. For double-heptamer, another round of 3D classification (K = 6) with particle alignment skipped was performed to improve the resolution. Finally, 153,005 particles were kept for the final 3D refinement (final resolution 6.8 Å). Due to the inter-heptamer flexibility, mask-based (only including a single heptamer) classification (K = 6) and refinement were applied, which showed no significant improvement on the maps (S3 Fig). In addition, multiple rounds of local classification focusing on the NBD ring or partial CLPB subunits, applied on the symmetry expansion derived half-particles, were also tested. However, none of them was able to reach better than 5 Å resolution.

For the sample of CLPB$^{E425Q}$-ATP, micrographs were acquired using a 300-kV FEI Titan Krios microscope (Gatan GIF K2 camera) at a nominal magnification of 36,000× (calibrated pixel size of 1.052 Å). A total of 32 frames with an exposure time of 8 s were collected with defocus ranged from −1.2 to −1.8 μm. A total of 4,574 micrographs were collected automatically using SerialEM [57]. After 2D classification and several round 3D classifications, a 7.5-Å

density map was achieved (S4 Fig). In addition, a density map of 5.2 Å could be achieved by applying a local mask of the NBD ring during reconstruction (S4C, S4D, and S4F Fig).

For the sample of CLPB<sup>isoform1</sup>-AMPPNP, a total of 953 micrographs were collected using a 200-kV FEI Talos Arctica microscope (Gatan K2 camera) at a nominal magnification of 36,000× (calibrated pixel size of 1.157 Å), and 32 frames with exposure time of 8 s were collected with defocus ranged from −1.1 to −1.8 μm. After several rounds of 2D classifications, only hexameric rings of top views were observed (S5C Fig).

For the sample of CLPB-casein-ATPγS, 20 μL casein (4.0 mg/mL) was added into 40 μL CLPB (2.0 mg/mL) before vitrification. A total of 885 micrographs were recorded with 200-kV FEI Talos Arctica (Gatan K2 camera) at a magnification of 36,000× (calibrated pixel size of 1.157 Å). A total of 32 frames with exposure time of 8 s were collected with defocus ranged from −1.4 to −2.0 μm under low-dose condition. After several rounds of 2D classifications, 54,771 particles were left, and hexameric rings (25,867 particles) and heptameric rings (24,046 particles) of top views were both observed (S5D Fig). As a control, CLPB-ATPγS dataset was prepared as CLPB-casein-ATPγS without adding casein.

For the sample of NBD<sup>E425Q</sup>-ATP, micrographs were recorded with 300-kV FEI Titan Krios G3i microscope (Gatan GIF K3 camera) at a nominal magnification of 64,000× (pixel size of 1.08 Å). A total of 32 frames with exposure time of 2.44 s were collected with defocus ranged from −1 to −1.4 μm. A total of 3,453 micrographs were collected, and 3,753,541 auto-picked particles were split into 2 parts to facilitate processing. After several rounds of 2D and 3D classification, 3 different major states were found: hexamer, heptamer, and nonamer. The density map of hexamer particles was refined to 7.4 Å (S10 Fig). The particle sets of heptamer and nonamer were further optimized, and after CTF refinement and Bayesian polishing, the final density maps were determined at resolutions of 4.1 Å and 3.7 Å, respectively (S10 Fig).

For the sample of NBD<sup>WT</sup>-AMPPNP, a total of 720 micrographs were recorded with 200-kV FEI Talos Arctica microscope (Gatan K2 camera) at a nominal magnification of 36,000× (calibrated pixel size of 1.157 Å), and 32 frames with exposure time of 8 s were collected with defocus ranged from −1.2 to −2.0 μm. After 2D classification, both hexameric rings and heptameric rings of top views were observed (S9 Fig).

The parameters and statistics for data collection and processing were summarized in S1 Table.

## Crystallization, data collection, and structure determination of the ANK domain

Plate-shaped crystals of CLPB-ANK were obtained within 3 weeks by sitting drop vapor diffusion method at 16°C. Typically, a volume of 1 μL of protein solution at a concentration of 6 to 12 mg/mL in the GF buffer (5 mM Tris-HCl (pH 7.5), 100 mM NaCl) was mixed with an equal volume of a precipitant well solution of 0.2 M ammonium citrate tribasic (pH 7.0), 20% (w/v) polyethylene glycol 3,350. Crystals were directly frozen and stored in liquid nitrogen prior to data collection. The final dataset was collected at the Shanghai Synchrotron Radiation Facility (SSRF) beamline BL18U1 (wavelength = 0.97915 Å, temperature = 100 K). A 900 diffraction images were collected with oscillation step of 0.2°. The data were merged and scaled using XDS [58] and Aimless [59]. The initial structure solution was obtained using the molecular replacement program PHASER [60] followed by AutoBuild [61] with a prediction structure of CLPB by AlphaFold2 [22]. Further model building was done using Coot and Phenix [62,63]. Data collection and refinement statistics are summarized in S2 Table and the atomic coordinates and structure factors have been deposited in the protein data bank (PDB ID: 7XC5).

## Model building and refinement

Model building was based on the predicted initial model of CLPB by AlphaFold2 [22] and the crystal structure of the ANK domain. The initial models were first docked into the cryo-EM density map using UCSF Chimera [64], rebuilt manually in Coot and refined (real-space) using Phenix [62,63]. Figures preparation and structure analysis were performed with UCSF ChimeraX [65] and Chimera [64].

## Supporting information

**S1 Fig. Biochemical and functional characterization of CLPB proteins.** (**A**) The purification of CLPB-N92, analyzing by size-exclusion chromatography (left) and SDS-PAGE (right). (**B**) Negative staining electron microscopy of the peak fraction in (A). Result shows that CLPB-N92 formed large aggregates and was highly heterogenous in size. (**C**) Time course of the C-terminal Step-tagged full-length CLPB (CLPB-FL) expression in transiently transfected HEK-293T cells. (**D**) CLPB-FL, CLPB-N92, and CLPB-N127 expression in transiently transfected HEK-293T cells. Data show that the CLPB-N127 has the same molecular weight as the mature form of CLPB. (**E**) ATPase activity of CLPB-N127 (S1 Data). (**F**) Disaggregase activity of CLPB-N127 (S1 Data).
(TIF)

**S2 Fig. Sample preparation of CLPB-AMPPNP, CLPB[E425Q]-ATP, and CLPB[isoform1]-AMPPNP.** (**A**) Purification of CLPB using size-exclusion chromatography with AMPPNP (red line) or without AMPPNP (black line). Corresponding fractions were analyzed by SDS-PAGE (right panel). (**B**) Representative nsEM image of the peak fraction in (A). (**C**) Purification of CLPB[E425Q] using size-exclusion chromatography in the presence of ATP. Corresponding fractions were analyzed by SDS-PAGE (right panel). (**D**) Representative nsEM image of the peak fraction in (C). (**E**) Purification of CLPB[isoform1] using size-exclusion chromatography. Corresponding fractions were analyzed by SDS-PAGE (right panel). (**F**) Representative nsEM image of the peak fraction in (E).
(TIF)

**S3 Fig. Image processing workflow of the CLPB-AMPPNP dataset.** (**A**) Image processing workflow of the CLPB dataset (see Methods for details). (**B, C**) Fourier shell correlation (FSC) curves for the final cryo-EM map of the double-heptameric complex (B) or heptameric complex (C) using the gold standard FSC 0.143 criteria.
(TIF)

**S4 Fig. Image processing workflow of the CLPB[E425Q]-ATP dataset.** (**A**) Image processing workflow of the CLPB[E425Q] dataset. (**B, C**) Local resolution estimation of CLPB[E425Q] double-hexamer (B) or NBD alone (C). (**D**) Density map of the NBD alone. The central substrate is colored yellow. (**E, F**) Fourier shell correlation curve of the final map of the CLPB[E425Q] double-hexamer (E) and NBD (F).
(TIF)

**S5 Fig. Conversion of the CLPB double-heptamer to double-hexamer upon substrate binding.** (**A–D**) Representative 2D classification averages of CLPB (A), CLPB[E425Q] (B), CLPB[isoform1] (C), and CLPB+Casein (D) datasets. The top views with hexameric features are indicated by red boxes.
(TIF)

**S6 Fig. The ANK domain is essential for the disaggregase activity of CLPB.** (**A**) SDS-PAGE analysis of the purified proteins. (**B**) ATPase assays of CLPB, NBD, and NBD[E425Q]. Results show that NBD[E425Q] has nearly no ATPase activity. NBD has strong ATPase activity. ATPase

activity was compared to CLPB ($N$ = 3, individual data points shown as dots, bars show mean ± SD, $^*p < 0.05$, $^{**}p < 0.01$, $^{***}p < 0.0001$) (S3 Data). (**C**) Disaggregase activity assay of CLPB, NBD, and NBD$^{E425Q}$. The results show that NBD and NBD$^{E425Q}$ abolish the disaggregase activity of CLPB. Disaggregase activity was compared to CLPB ($N$ = 3, individual data points shown as dots, bars show mean ± SD, $^{**}p < 0.01$, $^{***}p < 0.0001$) (S3 Data).
(TIF)

**S7 Fig. The unique insertion in ANK domain mediates higher-order structure formation and directly contact the substrate.** (**A**) Residual densities around the ANK domains, extending towards the central channel of the CLPB$^{E425Q}$ complex in the substrate-bound state. The extra densities are highlighted by a dotted ellipse. The density of the substrate is shown in yellow. (**B**) SDS-PAGE analysis of the purified proteins. (**C–E**) Purification of CLPB, CLPB$^{\Delta201–232}$, and CLPB$^{isoform1}$ complexes using size-exclusion chromatography.
(TIF)

**S8 Fig. The ANK domain dimer interface in the double-heptamer and double-hexamer.** (**A, B**) Rigid-body fitting of the crystal structure of the ANK domain into the density maps of the double-heptamer (A) and double-hexamer (B). (**C**) Purification of the CLPB, CLPB$^{R178E}$, and CLPB$^{R227E}$ complexes using size-exclusion chromatography. (**D**) SDS-PAGE analysis of the purified proteins. (**E**) Representative nsEM images of the CLPB (black line rectangle), CLPB$^{R178E}$ (blue line rectangle), and CLPB$^{R227E}$ complexes (red line rectangle).
(TIF)

**S9 Fig. Sample preparation of the NBD$^{E425Q}$-ATP and NBD-AMPPNP complex.** (**A**) Purification of NBD$^{E425Q}$ using size-exclusion chromatography in the presence of ATP. Corresponding fractions were analyzed by SDS-PAGE. (**B, C**) Representative nsEM image (B) and 2D classification averages of cryo-EM dataset (C) of the peak fraction in (A). (**D**) Purification of NBD using size-exclusion chromatography in the presence of AMPPNP. Corresponding fractions were analyzed by SDS-PAGE (right panel). (**E, F**) Representative nsEM image (E) and 2D classification averages cryo-EM dataset (F) of the peak fraction in (D). The hexameric ring is more compact than heptameric ring, and the diameter of central pore of hexamer is much smaller than that of heptamer.
(TIF)

**S10 Fig. Image processing workflow of the NBD$^{E425Q}$ dataset.** (**A**) Image processing workflow of the NBD$^{E425Q}$ dataset. The micrographs were subjected to motion correction and CTF estimation. The auto-picked particles were subjected to multiple rounds of 2D and 3D classifications. Three different oligomeric arrangements, hexameric, heptameric, and nonameric were identified. (**B–D**) Local resolution estimation of the density maps of the hexamer (B), heptamer (C), and nonamer (D) in (A). (**E**) Fourier shell correlation (FSC) curves of the final cryo-EM maps of hexamer (blue line), heptamer (black line), and nonamer (red line), using the gold standard FSC 0.143 criteria.
(TIF)

**S11 Fig. Nucleotide-binding states of the ATPase sites in the nonamer.** All the 8 ATPase sites in the nonamer are occupied by ATP. The atomic models are color-coded for different protomers. The segmented density maps of ATP were superimposed with the atomic model.
(TIF)

**S1 Table. Cryo-EM data collection, refinement, and validation statistics.**
(DOCX)

**S2 Table. Data collection and refinement statistics of the CLPB_ANK.**
(DOCX)

**S3 Table. The 24 enriched mitochondrial proteins highlighted in red in Fig 6B.**
(XLSX)

**S4 Table. Top 20 abundant and highly enriched mitochondrial proteins located in the OM, IMS, and IM.**
(XLSX)

**S1 Raw image. Uncropped western blotting and SDS-PAGE gel images in this work.**
(PDF)

**S1 Data. The ATPase activity and disaggregase activity of CLPB-N127.**
(XLSX)

**S2 Data. The proportion of the hexameric and heptameric top views in the CLPB-AMPPNP, CLPBE425Q-ATP, CLPBisoform1-AMPPNP, CLPB-ATPγs, and CLPB +Casein-ATPγs datasets.**
(XLSX)

**S3 Data. The ATPase activity and disaggregase activity of CLPB, NBD, and NBD$^{E425Q}$.**
(XLSX)

**S4 Data. The ATPase activity and disaggregase activity of CLPB, CLPB$^{\Delta 201-232}$, CLPB$^{isoform1}$ CLPB$^{R178E}$, and CLPB$^{R227E}$.**
(XLSX)

**S5 Data. Differential enrichment profile of mitochondrial proteins in the Strep/HA tagged ANK samples.** Mitochondrial proteins were analyzed based on a stringent screening condition ($p$-value < 0.01 and Fold change > 4).
(XLSX)

**S6 Data. Enrichment analysis of mitochondrial proteins with OM, IMS, and IM location in the ANK and ANK-isoform1 samples.**
(XLSX)

**S7 Data. Enrichment analysis of mitochondrial proteins with OM, IMS, and IM location in the ANK and ANK-Δloop samples.**
(XLSX)

## Acknowledgments

We thank the Core Facilities at the School of Life Sciences, Peking University for help with negative staining EM; the Electron Microscopy Laboratory and Cryo-EM Platform for help with data collection; the High-performance Computing Platform for help with computation; and the National Centre for Protein Sciences at Peking University for assistance with MS.

## Author Contributions

**Funding acquisition:** Ning Gao.

**Investigation:** Damu Wu, Yan Liu, Yuhao Dai, Guopeng Wang, Guoliang Lu, Yan Chen, Ningning Li, Jinzhong Lin, Ning Gao.

**Supervision:** Ning Gao.

**Writing – original draft:** Ning Gao.

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
