## [Editor Report · Decision Letter 0]

13 May 2022

Dear Dr gao, 

Thank you for submitting your manuscript entitled "Unique Structural and Functional Properties of a Mitochondrial AAA+ Disaggregase CLPB" for consideration as a Research Article by PLOS Biology. Please accept my apologies for the delay in getting back to you as we consulted with an academic editor about your submission.

Your manuscript has now been evaluated by the PLOS Biology editorial staff, as well as by an academic editor with the relevant expertise, and I am writing to let you know that we would like to send your submission out for external peer review.

Once your full submission is complete, your paper will undergo a series of checks in preparation for peer review. Once your manuscript has passed the checks it will be sent out for review. To provide the metadata for your submission, please Login to Editorial Manager (https://www.editorialmanager.com/pbiology) within two working days, i.e. by May 15 2022 11:59PM.

If your manuscript has been previously reviewed at another journal, PLOS Biology is willing to work with those reviews in order to avoid re-starting the process. Submission of the previous reviews is entirely optional and our ability to use them effectively will depend on the willingness of the previous journal to confirm the content of the reports and share the reviewer identities. Please note that we reserve the right to invite additional reviewers if we consider that additional/independent reviewers are needed, although we aim to avoid this as far as possible. In our experience, working with previous reviews does save time. 

If you would like to send previous reviewer reports to us, please email me at rhodge@plos.org to let me know, including the name of the previous journal and the manuscript ID the study was given, as well as attaching a point-by-point response to reviewers that details how you have or plan to address the reviewers' concerns. 

Kind regards,

Richard

Richard Hodge, PhD

Associate Editor, PLOS Biology

rhodge@plos.org

PLOS

---

## [Decision Letter · Decision Letter 1]

30 Jun 2022

Dear Dr gao,

Thank you for your patience while your manuscript "Unique Structural and Functional Properties of a Mitochondrial AAA+ Disaggregase CLPB" was peer-reviewed at PLOS Biology. Please accept my sincere apologies for the great delay in getting back to you with our decision. It has now been evaluated by the PLOS Biology editors, an Academic Editor with relevant expertise, and by four independent reviewers. 

As you will see, the reviewers find your manuscript interesting and worth pursuing, however they also raise several points that will need to be addressed in order for us to consider publication of the manuscript. Reviewer #3 raises concerns that the double-heptameric and nonameric structures could be a structural artefact and asks that independent biochemical validation data is provided. In addition, the reviewers ask for further clarifications since several pieces of data contrast with previously published work, as well as raising overlapping concerns with the interpretation of the data and the proposed ANK domain binding model.

In light of the reviews, which you will find at the end of this email, we would like to invite you to revise the work to thoroughly address the reviewers' reports. Given the extent of revision needed, we cannot make a decision about publication until we have seen the revised manuscript and your response to the reviewers' comments. Your revised manuscript is likely to be sent for further evaluation by all or a subset of the reviewers.

**IMPORTANT - SUBMITTING YOUR REVISION**

*Re-submission Checklist*

*Published Peer Review*

*PLOS Data Policy*

*Blot and Gel Data Policy*

Sincerely,

Richard

Richard Hodge, PhD

Associate Editor, PLOS Biology

rhodge@plos.org

REVIEWS:

Reviewer #1: The human mitochondrial AAA+ ATPase CLPB (aka, HSP78 and Skd3)　is attracted attentions because it can mediate disaggregation of various proteins and also is relaled with various diseases. Compared with canonical AAA+, is unique in the N terminal ankyrin repeat domain (ANK).

The manuscript describes the cryoEM structures of CLPB and studied the role of the ANK domain.

The authors found that CLPB assembled into homo-tetradecamers in apo-state and changed to homo-dodecamers upon binding to substrates. As the ANK domain located at the interface of heptamer or hexamer, the authors concluded that the ANK domain is responsible for the higher-order assembly. 

The authors also suggested that the ANK domain is substrate binding site from the results of MS analysis of ANK interactomes.

The putative disaggregases, Hsp104 or ClpB, contains two NBD in a subunit. Thus it seems to be reasonable that CLPB with one NBD exists as double ring complex to take similar configuration.

The experiments were well designed and the results are persuasive. Therefore, the manuscript is valid for publication. However, I recommend the authors to consider and discuss a couple of following problems raised from the results.

The transition from tetradecamer to dodecamer requires disassembly of the complex. Thus, CLPB should take dynamic oligomeric structural chanse, possible with the dissociation to monomers. The relatively low resolution should be due to the dynamic structural change.

The ANK domain exists in the interface between the rings. Thus, it seems to be difficult to contribute to the interaction with substrates. I think the interactome analysi of the ANK domain might be artefact.

I think that the manuscript is acceptable after the revision according to the above problems. 

Reviewer #2: In this study, Wu et al. present a structural characterization of the human HSP100 family protein CLPB. Although diverse conformations of the purified protein limited the resolution, the authors used rigid body fitting to model the CLPB structure. Interestingly, the authors found that CLPB assembles into homo-tetradecamers in the apo-state which are converted into dodecamers upon substrate binding. The study confirms the importance of ANK domain for the substrate recognition and suggests its importance for higher-order organization (double hexameric structure) that is critical for disaggregase cycle of CLPB.

This is interesting study which provides novel and unexpected insights into the structural organization of CLPB and which allows mapping and functional interpretation of disease-related mutations and understanding their possible implications on the CLPB function. However, the authors should consider the following points:

1. The authors refer to substrate density in the central channel of CLPB hexamers in Fig. 1D. However, it remains unclear how a substrate density is distinguished from conformational changes in CLPB that are induced by the Walker B mutation (impairing ATP hydrolysis)? Why was AMP-PNP added in these experiments? The experiments shown in Fig. 2C appears critical in this context, but it remains unclear from the description in the legend whether or not ATPyS was present in all samples. Why was ATPyS rather than AMP-PNP added? Is the hexameric state induced upon substrate binding or only upon substrate binding in the presence of ATPyS? These experiments are central as they aim to establish a substrate induced change in the stoichiometry of CLPB complexes, a central and perhaps the most intriguing finding of the manuscript.

2. The mass spectrometric analysis of CLPB interacting protein shown in Fig. 6 is not sufficiently documented and therefore difficult to assess. I could not find any description detailing the MS analysis of the data (number of biological replicates, statistical evaluation, etc). If I understood correctly, MTS-ANK-HA was used as a control in this experiment. As the authors found that the deletion of the AA 201-232 in the ANK domain significantly reduces CLPB disaggregase activity and that this part of the protein might be involved in substrate recognition and transport of the substrate to AAA+ domain, such mutant (MTS-ANK�201-232Strep) would be great negative control in a search for the specific CLPB substrates. Moreover, the experiments were seemingly performed in the presence of endogenous CLPB? Is endogenous CLPB found in the eluate? Experiments in CLPB knock-out cells would more informative.

3. The authors report that ANK motif deletion does not perturb ATPase activity (or even see higher ATPase activity of the NBD mutant). This result is in contrast to the published result by Cupo and Shorter that report lack of ATPase activity of the isolated NBD (consistent with a similar lack of activity of isolated NBD2 from yeast Hsp104 or bacterial ClpB). How do authors explain this discrepant result?

Reviewer #3: Human CLPB (aka SKD3) is a protein disaggregase found in the mitochondrial intermembrane space, which promotes the solubilization of aggregated mitochondrial proteins (1, 2). CLPB consists of an ANK domain followed by a single AAA domain. It was previously shown that the ANK domain is essential to CLPB function (1) and missense mutations throughout the CLPB gene are associated with 3-methylglutaconic aciduria (3-MGA) and severe congenital neutropenia (SCN). Here, Wu et al. presents the cryo-EM structures of CLPB in the apo and casein-bound form. The reported oligomeric structures range from tetradecamers (heptamers), dodecamers (hexamers to nonamers. Although the work is technically sound, there are several inconsistencies and stark differences with published work that must be addressed. The introduction is erroneous in many places, e.g. CLPB/SKD3 is NOT an Hsp78 homolog; the N-domain of yeast Hsp104 is dispensable for protein disaggregation and by extension substrate binding; the NBDs of Hsp100 do not elicit a power stroke. There is also substantial overlap with published work (e.g. Ref. 1) and the significance of the presented data are not always clear (e.g. Fig. EV3). On balance, while the authors present a fine structure, the paper is rather confusing and highly speculative at times (nonamer structure), which precludes publication in its current form.

Major

1. The authors present cryo-EM structures of apo CLPB (heptamer/tetradecamer) and substrate-bound CLPB (hexamer/dodecamer) at medium resolution (~6-7 Å resolution). However, the authors did not further capitalize on this finding, and how CLPB goes from a heptamer to a hexamer is unclear.

2. The proposed heptamer (or tetradecamer) structure of apo CLPB is intriguing and contrasts published work showing that CLPB forms a hexamer both without substrate (3, 4) and with substrate (4). How can this be reconciled?

3. Perhaps even more perplexing is the structural discrepancy between apo CLPB (heptamer/tetradecamer) and CLPB-WB mutant (hexamer/dodecamer) co-purified with substrate. By co-purification, do the authors imply an E. coli protein that co-purified with CLPBWB? What is the identity of the bound protein and why is the bound polypeptide not shown in Fig. 1? As stated, the logic of a co-purified substrate-bound CLPB-WB is a stretch.

4. Heptamers, incl. tetradecamers, have been reported for several AAA+ motors including Clp/Hsp100 (c.f. also extensive literature on p97), but are largely dismissed as structural artifacts. Cupo et al. (4) previously showed that heptamers are a minor species contrasting the data presented in Fig. 2C. To test whether a CLPB heptamer/tetradecamer is the predominant species, the authors need to independently verify their heptamer structure by site-specific cross-linking and gel analysis, or mass photometry.

5. In addition to the widely observed hexamer/heptamer structures, the authors also observed a highly speculative nonamer structure at 3.7-Å resolution. Although the authors state that the nonamer structure is non-physiological, it is important to validate any structural insight derived from an artifactual structure. To test whether the nonamer structure has any functional relevance, independent biochemical data using CLPB mutants must be presented. If not, the nonamer structure is confusing and does not add to the conclusion.

6. Hsp100 motors translocate substrate in a unidirectional manner and bind substrates via their N-domains (here: ANK domain). In the present structure, CLPB adopts a head-to-head oligomer held together by the ANK domains, which precludes a substrate interaction via the CLPB N-domain. Since the ANK domains mediate multiple interactions with neighboring subunits both within one oligomer and across oligomers, how do they engage with substrate (not to mention large aggregates)?

7. The mass spectrometry data presented in Fig 6C is interesting and largely consistent with published work (1, 5). However, according to MitoCarta 3.0, HAX-1 is not an IMS protein and may not even be a mitochondrial protein at all. A discussion is required.

8. There are several erroneous statements in the introduction that need to be corrected. For instance, CLPB is a single-ring oligomer while ClpB (as well as Hsp78 and Hsp104) forms a two-ring hexamer. In essence human CLPB and microbial ClpB/Hsp104/Hsp78 belong to different Hsp100 classes. Hence, the term "human CLPB" is a misnomer since the protein is both structurally and functionally unrelated to bacterial ClpB. A re-review of the revised manuscript is imperative to ensure that statements made are factually correct.

Minor 

1. The Li et al., 2006 reference to support "…first protein…to contain both ANK domain and AAA+ activity" seems misplaced.

2. A Glu to Gln mutation of the Walker B does not necessarily abrogate ATP hydrolysis since a Gln is fully compatible with a nucleophilic attack. Independent biochemically verification is required.

3. Fig. EV1: It is widely appreciated that AMP-PNP introduces a structural artifact in Hsp100 proteins and is incompatible with the true "ATP-activated" state required for substrate binding. The Glu to Gln mutation in the Walker B motif is more physiological but does not necessarily abrogate ATP hydrolysis.

4. Data presented in Fig. EV3 are largely identical to published work (1).

5. Please use line and page numbers in any revision.

References

Cupo RR, Shorter J. Skd3 (human CLPB) is a potent mitochondrial protein disaggregase that is inactivated by 3-methylglutaconic aciduria-linked mutations. eLife. 2020;9:e55279.

Mróz D, Wyszkowski H, Szablewski T, Zawieracz K, Dutkiewicz R, Bury K, et al. CLPB (caseinolytic peptidase B homolog), the first mitochondrial protein refoldase associated with human disease. Biochim Biophys Acta. 2020;1864(4):129512.

Spaulding Z, Thevarajan I, Schrag LG, Zubcevic L, Zolkiewska A, Zolkiewski M. Human mitochondrial AAA+ ATPase SKD3/CLPB assembles into nucleotide-stabilized dodecamers. Biochem Biophys Res Commun. 2022;602:21-6.

Cupo RR, Rizo AN, Braun GA, Tse E, Chuang E, Southworth DR, et al. Unique structural features govern the activity of a human mitochondrial AAA+ disaggregase, Skd3. bioRxiv. 2022:2022.02.17.480866.

Chen X, Glytsou C, Zhou H, Narang S, Reyna DE, Lopez A, et al. Targeting mitochondrial structure sensitizes acute myeloid leukemia to venetoclax treatment. Cancer Discov. 2019;9(7):890-909.

Reviewer #4 (Christopher Hill, signs review): Structural and biochemical studies are reported of CLPB, a mitochondrial AAA disaggregase that is linked to human disease and is unusual among AAA unfoldases in having an N-terminal ANK domain.

Curiously, it is found that the ANK domain mediates face-to-face stacking of AAA ring structures, such that the active state of CLPB is proposed to be a dodecamer of two hexameric rings facing each other rather than the single hexameric ring envisioned for other AAA unfoldases. In the absence of bound substrate, CLPB is seen to form a tetradecamer in which two heptameric rings face each other in a configuration that is envisioned to be inactive.

Structures were also determined of the NBD lacking the ANK domain, and seen to bind substrate in hexameric, heptameric, and nonameric forms. Although the nonamer is envisioned to be not relevant physiologically, it gave the best structure and view of substrate interactions at 3.7Å resolution.

I do not have any technical concerns.

There are numerous instances where the use of English should be improved.

I am doubtful that the precision implied by the description of hydrogen bonding interactions is justified by the resolution of the reconstructions.

I am confused about the axial dimensions in Figure 4. Given that the substrate is modeled in an extended/beta conformation with two residues and 60° rotation per subunit, the axial displacement per subunit (24/6, 29/6, 32/6; 4-5.3Å/subunit) all seem too short. Given that the axial displacement is almost parallel to the substrate strand, I would expect axial displacements in the range of 6-7Å per subunit.

The authors propose that two hexamers may act on substrate in a coordinated manner within the dodecameric assembly. It is not clear to me, however, that an alternative and more conventional model has been disproven - specifically, that the ANK domains function to localize CLPB to substrate, perhaps by binding extended portions of protein aggregates, while single hexameric rings translocate/unfold the substrate. The authors may want to modify the discussion to acknowledge this possibility.

---

## [Decision Letter · Decision Letter 2]

26 Oct 2022

Dear Dr gao,

Thank you for your patience while we considered your revised manuscript "Unique Structural and Functional Properties of a Mitochondrial AAA+ Disaggregase CLPB" for publication as a Research Article at PLOS Biology. Please accept my sincere apologies for the long delays that you have experienced during this round of the peer review process. Your revised study has been evaluated by the PLOS Biology editors, the Academic Editor and the original reviewers.

The reviews are attached below. As you can see, whilst Reviewers #1, #2 and #4 are now satisfied with the revision, Reviewer #3 still raises outstanding concerns with the physiological relevance of the findings given the lack of biochemical validation of the double-heptamer structure, as well as the overall level of mechanistic insight into how the two CLPB subunits are ejected upon substrate binding during the transition between the double-hexamer and double-heptamer states.

After discussing the reviews at length with the Academic Editor, we feel that these experiments would improve the manuscript and are important to provide direct support for the existence of the ClpB oligomeric structures and the substrate binding model. Reviewer #3 also notes that a related paper, recently published in Cell Reports, has solved the double-hexamer ClpB structure. We ask that you please discuss and contextualize these findings in the revised manuscript. Please be assured that we will not see this paper as a precedent when considering your paper for publication. 

Given the extent of revision needed, we cannot make a decision about publication until we have seen the revised manuscript and your response to the reviewers' comments. Your revised manuscript is likely to be sent for further evaluation by all or a subset of the reviewers.

We expect to receive your revised manuscript within 3 months. Please email us (plosbiology@plos.org) if you have any questions or concerns, or would like to request an extension. At this stage, your manuscript remains formally under active consideration at our journal; please notify us by email if you do not intend to submit a revision so that we may withdraw it.

**IMPORTANT - SUBMITTING YOUR REVISION**

*Re-submission Checklist*

*Published Peer Review*

*PLOS Data Policy*

*Blot and Gel Data Policy*

Sincerely,

Richard

Richard Hodge, PhD

Associate Editor, PLOS Biology

rhodge@plos.org

REVIEWS:

Reviewer #1 (Masafumi Yohda, signs review): The authors answered to all queries from the reviewers and revised the manuscript according to the suggestions. Thus, I recommend this manuscript for publication.

Reviewer #2 (Thomas Langer, signs review): The authors have carefully considered points of criticisms on their original manuscript and significantly improved their manuscript. The newly added MS data on interaction partners of ANK domains and mutants thereof are convincing and strengthen the conclusion of the authors. This is an interesting manuscript providing new insight as to how CLPB handles its substrate, which will be of broad interest.

Reviewer #3: Wu et al reports the cryoEM structures of apo CLPB (double-heptamer) and ATP-bound CLPB with a bacterial substrate mimic (double-hexamer) at medium resolution. Based on these findings, the authors propose that the double-heptamer may represent a resting state, which is a new finding. The 3D structure of the double-hexamer with casein was recently published by Cupo et al Cell Rep, 2022. Although, both structures consist of a double-hexamer, Cupo et al favors that the active form is a hexamer consistent with other single-ring Hsp100 unfoldases, contrasting the present work.

A shortcoming of the present study is the lack of new mechanistic insight since both the 3D structure and MS data have been published by others. It is not so much that different oligomer states are observed without and with polypeptide, but how CLPB goes from a double-heptamer to a double-hexamer, which is of interest to the field. The authors propose that two CLPB subunits are "ejected" upon substrate binding, which seems implausible considering the double-ring structure. Furthermore, flexibility at the seam is very different from removal of CLPB subunits that are held within the ring complex and across rings via the ANK domain. While the presented cryoEM structures are solid, they appear to be non-physiological, of medium resolution, and more geared towards the specialist.

* Fig 2B and Rebuttal Fig 3 suggest that only CLPB(E425Q) with ATP forms a double-hexamer. ATP analogs give rise primarily to heptamers with casein driving the formation of hexamers. The latter is consistent with the published work by Cupo et al., 2022.; who only observed double-hexamers. No two 3D reconstructions are alike and whatever the correct answer might be, according to the present study, it would seem that the CLPB structures with AMP-PNP and ATPγS represent off-pathway states.

* Heptamer structures were observed with different AAA+ proteins, often at low resolution, and frequently by cryoEM, and are largely dismissed as an in vitro artifact. The fact that heptamers and other oligomeric species are being reported in the literature (lines 292-294) does not make them physiological. Of note, a bacterial homolog of Bcs1 was recently shown to be a hexamer. The authors stated that the nonamer is non-physiological and no follow-up mutational work was done. Consequently, the value the nonamer structure is unclear and undoubtedly will further clutter the literature.

* Line 106-107: As a protein disaggregase, CLPB recognizes aggregated protein as native substrate. Only because a Walker B CLPB mutant can bind an endogenous bacterial protein during protein expression does not imply that a substrate complex was obtained. Independent biochemical verification is required. Since mitochondrial CLPB is exclusively found in animal cells, casein may actually be a more physiological substrate mimic.

* Assuming that the double-hexamer structure is physiological, is it possible that a substrate is threaded from the NBD end? Since the authors captured an unknown bacterial protein during protein expression, one might expect to see additional mass density that could indicate the direction of the polypeptide bound in the axial channel.

* The authors claim that the ANK domain has no lateral interactions in the double-hexamer/heptamer (c.f. response to reviewers). Yet, Fig 3G, clearly shows lateral contacts that are also described in the text (lines 182-194). The observation that mutating R178 and/or R227 to Glu severely impaired protein disaggregation is very interesting. Both Arg are part of the lateral interactions between neighboring ANK domains but do not abolish the double hexamer/heptamer when mutated. Since R178 or R227 are not involved in mediating an inter-subunit ANK interaction nor can directly interact with substrate for steric reasons, it would argue that the double-heptamer/hexamer must dissociate and therefore is an in vitro artifact.

* According to MitoCarta 3.0 (1), none of the 20 highest scoring CLPB substrates listed in "S5-HA-Strep_top 20" are localized to the IMS. This includes OPA1 and AGK, which is surprising. The authors proposed that OPA1 is a CLPB substrate that binds to the double-heptamer to induce the structure of the double-hexamer complex with substrate. OPA1 is a protein of 111-KDa and forms large, multi-subunit tubules (2). Even if CLPB recognizes monomeric OPA1 in non-aggregated form, the protein is too large to "fit into" the double-heptamer without dissociating the oligomer.

* While several studies suggested that HAX1 is a CLPB substrate (3, 4), HAX-1 is not a mitochondrial protein (1, 5). We further note that neither PRKD2 nor HSP27 (1), whose solubility is apparently controlled by HAX1 in mitochondria (4), are mitochondrial proteins. This work is cited by the authors as evidence for a direct HAX1:CLPB interaction.

* Fig. S2A and S2E: The SEC chromatogram shows two peaks for CLPB-AMPPNP and CLPBiso1-AMPPNP of near equal height (S2E). What is the nature of the second species consisting of fractions: 16-22, which were not included in the SDS-PAGE gel?

* Line 305: "potentially thread the substrate in both directions, similar as bacterial ClpX and ClpA". This is a misleading statement. Unlike CLPB, bacterial ClpX and ClpA are stably bound to the ClpP peptidase that confers unidirectional substrate-threading from an N-to-C direction. However, this may not be the case for CLPPB.

Responses to reviewers' queries: 

* (Rev 1-3): Obviously there are lateral interactions between ANK domains of neighboring subunits (c.f. Fig 3G). It is unclear what is implied by stating that "[the ANK domain] has no lateral interaction with the adjacent ANK domains within the hexamer/heptamer.

* (Rev. 2): ATP, AMP-PNP, and ATPgS are quite different and sometimes promote off-pathway conformations. They also perform differently is substrate interaction. Structurally they are quite different as well, which can be seen at atomic resolution but is beyond the resolution of the present cryoEM reconstructions. The rebuttal figure 2 is unsatisfactory because Hsp104 is an entirely different protein. The authors should perform this experiment with their CLPB protein used for structure analysis.

* (Rev. 3): "a very recent NMR study on Hsp104 has identified a substrate-binding groove in the NTD to recognize exposed hydrophobic stretches in substrates of Hsp104 (Harari et al., 2022)). The ANK domain may work in a similar fashion to bind to exposed hydrophobic sequences of the substrate." The NTD of Hsp104 and CLPB are neither related nor similar in structure. Inferring any mechanistic insights by direct comparison is wrong.

* (Rev 3): "a study employed proteinase K assay on isolated mitochondria and showed that the degradation of HAX1 (CLPB as well) requires the disruption of the outer membrane (Fan et al., 2022). Therefore, we think that it could be a mis-annotation of MitoCarta 3.0." I do not doubt that HAX-1 may mimic a substrate (as casein does), but that does not mean this interaction is physiological and needs to be taken with caution.

1. Rath S, Sharma R, Gupta R, Ast T, Chan C, Durham TJ, et al. MitoCarta3.0: an updated mitochondrial proteome now with sub-organelle localization and pathway annotations. Nucleic Acids Res. 2021;49(D1):D1541-d7.

2. Zhang D, Zhang Y, Ma J, Zhu C, Niu T, Chen W, et al. Cryo-EM structures of S-OPA1 reveal its interactions with membrane and changes upon nucleotide binding. Elife. 2020;9.

3. Chen X, Glytsou C, Zhou H, Narang S, Reyna DE, Lopez A, et al. Targeting mitochondrial structure sensitizes acute myeloid leukemia to venetoclax treatment. Cancer Discov. 2019;9(7):890-909.

4. Fan Y, Murgia M, Linder MI, Mizoguchi Y, Wang C, Łyszkiewicz M, et al. HAX1-dependent control of mitochondrial proteostasis governs neutrophil granulocyte differentiation. J Clin Invest. 2022;132:e153153.

5. Jeyaraju DV, Cisbani G, De Brito OM, Koonin EV, Pellegrini L. Hax1 lacks BH modules and is peripherally associated to heavy membranes: implications for Omi/HtrA2 and PARL activity in the regulation of mitochondrial stress and apoptosis. Cell Death Differ. 2009;16:1622-9.

Reviewer #4 (Christopher Hill, signs review): The revised manuscript seems acceptable for publication. However, it would benefit substantially from copying editing to improve the use of English.

---

## [Editor Report · Decision Letter 3]

15 Dec 2022

Dear Ning,

Thank you for your patience while we considered your revised manuscript "Structural and functional insights into the action mode of a mitochondrial AAA+ disaggregase CLPB" for publication as a Research Article at PLOS Biology. This revised version of your manuscript has been evaluated by the PLOS Biology editors and the Academic Editor.

Based on our Academic Editor's assessment of your revision, we are likely to accept this manuscript for publication, provided you satisfactorily address the remaining data and other policy-related requests that I have provided below (A-G):

(A) We would like to suggest the following modification to the title:

“Comprehensive structural characterization of the human AAA+ disaggregase CLPB in the apo- and substrate-bound states reveals a unique mode of action driven by oligomerization"

(B) We ask that you please incorporate the following edits into the Abstract:

'The human AAA+ ATPase CLPB (SKD3) is a protein disaggregase in the mitochondrial intermembrane space and functions to promote the solubilization of various mitochondrial proteins. Loss-of-function CLPB mutations are associated with a few human diseases with neutropenia and neurological disorders. Unlike canonical AAA+ proteins, CLPB contains a unique ankyrin repeat domain (ANK) at its N-terminus. How CLPB functions as a disaggregase and the role of its ANK domain are currently unclear. Herein, we report a comprehensive structural characterization of human CLPB in both the apo- and substrate-bound states. CLPB assembles into homo-tetradecamers in apo-state and is remodeled into homo-dodecamers upon substrate binding. Conserved pore-loops on the ATPase domains form a spiral staircase to grip and translocate the substrate in a step-size of two amino acid residues. The ANK domain is not only responsible for maintaining the higher-order assembly but also essential for the disaggregase activity. Interactome analysis suggests that the ANK domain may directly interact with a variety of mitochondrial substrates. These results reveal unique properties of CLPB as a general disaggregase in mitochondria and highlight its potential as a target for the treatment of various mitochondria-related diseases.'

(C) You may be aware of the PLOS Data Policy, which requires that all data be made available without restriction: http://journals.plos.org/plosbiology/s/data-availability. For more information, please also see this editorial: http://dx.doi.org/10.1371/journal.pbio.1001797

- Supplementary files (e.g., excel). Please ensure that all data files are uploaded as 'Supporting Information' and are invariably referred to (in the manuscript, figure legends, and the Description field when uploading your files) using the following format verbatim: S1 Data, S2 Data, etc. Multiple panels of a single or even several figures can be included as multiple sheets in one excel file that is saved using exactly the following convention: S1_Data.xlsx (using an underscore).

 - Deposition in a publicly available repository. Please also provide the accession code or a reviewer link so that we may view your data before publication.

Figure 2B, 3C-F, 6B, 6E-F, S1E-F, S6B-C

(D) Thank you for depositing the structural data in the PDB and EMDB databases However, I note that the data is currently on hold. We ask that you please make this data publicly available before publication. 

(E) Please also ensure that each of the relevant figure legends in your manuscript include information on *WHERE THE UNDERLYING DATA CAN BE FOUND*, and ensure your supplemental data file/s has a legend.

(F) We require the original, uncropped and minimally adjusted images supporting all blot and gel results reported in the following Figures:

Figure 6A, 6D, S1A, S1C-D, S2A, S2C, S2E, S6A, S7B, S8D, S9A, S9D

We will require these files before a manuscript can be accepted so please prepare and upload them now. Please carefully read our guidelines for how to prepare and upload this data: https://journals.plos.org/plosbiology/s/figures#loc-blot-and-gel-reporting-requirements

(G) Please ensure that your Data Statement in the submission system accurately describes where your data can be found and is in final format, as it will be published as written there.

We expect to receive your revised manuscript within two weeks. 

*Published Peer Review History*

*Press*

Sincerely,

Richard

Richard Hodge, PhD

Associate Editor, PLOS Biology

rhodge@plos.org

PLOS

---

## [Editor Report · Decision Letter 4]

4 Jan 2023

Dear Ning,

Thank you for submitting your revised Research Article "Comprehensive structural characterization of the human AAA+ disaggregase CLPB in the apo- and substrate-bound states reveals a unique mode of action driven by oligomerization" for publication in PLOS Biology. Please accept my apologies for the delay due to the recent Christmas holiday period. 

On behalf of my colleagues and the Academic Editor, Ursula Jakob, I am pleased to say that we can accept your manuscript for publication, provided you address any remaining formatting and reporting issues. These will be detailed in an email you should receive within 2-3 business days from our colleagues in the journal operations team; no action is required from you until then. Please note that we will not be able to formally accept your manuscript and schedule it for publication until you have completed any requested changes.

PRESS

Best wishes,

Richard 

Richard Hodge, PhD

Associate Editor, PLOS Biology

rhodge@plos.org

PLOS
